# ReSpinQuant: Efficient Layer-Wise LLM Quantization via Subspace Residual Rotation Approximation

Suyoung Kim [1]  Sunghyun Wee [* 1 2]  Hyeonjin Kim [* 1]  Kyomin Hwang [* 1]  Hyunho Lee [* 1]  Nojun Kwak [1]

## Abstract

Rotation-based Post-Training Quantization (PTQ) has emerged as a promising solution for mitigating activation outliers in the quantization of Large Language Models (LLMs). Global rotation methods achieve inference efficiency by fusing activation rotations into attention and FFN blocks, but suffer from limited expressivity as they are constrained to use a single learnable rotation matrix across all layers. To tackle this, layer-wise transformation methods emerged, achieving superior accuracy through localized adaptation. However, layer-wise methods cannot fuse activation rotation matrices into weights, requiring online computations and causing significant overhead. In this paper, we propose **ReSpinQuant**, a quantization framework that resolves such overhead by leveraging offline activation rotation fusion and matching basis using efficient residual subspace rotation. This design reconciles the high expressivity of layer-wise adaptation with only negligible inference overhead. Extensive experiments on W4A4 and W3A3 quantization demonstrate that ReSpinQuant achieves state-of-the-art performance, outperforming global rotation methods and matching the accuracy of computationally expensive layer-wise methods with minimal overhead.

## 1. Introduction

As Large Language Models (LLMs) continue to scale, their deployment on resource-constrained devices faces significant challenges due to massive memory footprints and computational costs. Post-Training Quantization (PTQ) has emerged as a standard solution, compressing weights and

---
[*]Equal contribution [1]Seoul National University, Seoul, Republic of Korea [2]LG Electronics, Seoul, Republic of Korea. Correspondence to: Nojun Kwak <nojunk@snu.ac.kr>.

*Proceedings of the 43ʳᵈ International Conference on Machine Learning*, Seoul, South Korea. PMLR 306, 2026. Copyright 2026 by the author(s).

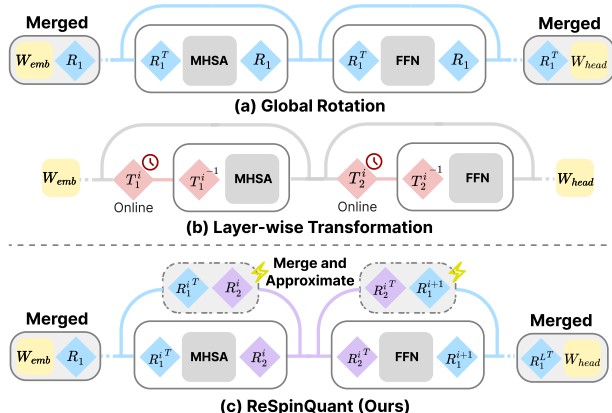

*Figure 1.* **Comparison of rotation paradigms.** (a) **Global Rotation** ensures efficiency via offline weight merging but limits expressivity. (b) **Layer-wise Transformation** enhances expressivity but incurs high online overhead. (c) **ReSpinQuant (Ours)** reconciles this trade-off by merging layer-wise rotation into weights, and resolves the basis mismatch problem via subspace residual rotation approximation, achieving both high accuracy and efficiency.

activations to low-bit precision (*e.g.*, 4-bit or 3-bit). However, the presence of extreme outliers in activation channels poses a critical bottleneck, as they disproportionately expand the dynamic range, leading to severe quantization errors (Dettmers et al., 2022; Xiao et al., 2023).

To address the outlier problem, rotation-based quantization methods, such as QuaRot (Ashkboos et al., 2024) and SpinQuant (Liu et al., 2025), have been proposed. These methods apply orthogonal transformations to the activation space, redistributing outliers across multiple dimensions to flatten the distribution. The paradigm can be categorized into two approaches: **1) Global Rotation** applies a consistent rotation across the entire model (Figure 1-(a)). This approach enables offline activation rotation fusion, keeping the inference overhead minimal by merging the activation rotation matrix into the weights. However, since all layers are forced to share a single basis, layer-specific outlier patterns cannot be handled individually. **2) Layer-wise Transformation** addresses this limitation by assigning a unique rotation matrix to each layer. While this strategy achieves better optimization with more parameters, layer-wise methods incur substantial computational overhead. As shown in Figure 1-(b),

it prevents activation rotations from being fused into weight matrices offline, enforcing additional online computations.

Recent work on layer-wise operations has proposed various structured approximations to mitigate the overhead. OS-TQuant (Hu et al., 2025) restricts the transformation to a scaling vector ($\mathcal{O}(D)$). ButterflyQuant (Xu et al., 2025) and ParoQuant (Liang et al., 2026) constrain the rotation structure to butterfly factors to reduce complexity to $\mathcal{O}(D \log D)$, while FlatQuant (Sun et al., 2025) employs Kronecker products with $\mathcal{O}(D^{1.5})$ complexity. While structured matrices are employed in these methods to mitigate complexity, they inevitably restrict the model's representational capacity.

In this paper, we present ReSpinQuant, a layer-wise rotation-based quantization method, which employs unconstrained matrices to unlock model capability without increasing computational overhead (Figure 1-(c)). Specifically, our empirical observations reveal that these learned rotation matrices do not deviate significantly from their initial state when initialized with a Hadamard matrix and optimized using the Cayley optimizer (Li et al., 2020). Consequently, the transition matrix for residual connections, $\mathbf{T} = \mathbf{R}_{in}^T \mathbf{R}_{out}$, remains mathematically close to the Identity matrix. Building on this insight, we propose approximating the computationally expensive dense transition $\mathbf{T}$ using subspace projection. Rather than performing a full $D \times D$ matrix multiplication, our method projects the residual into a low-dimensional subspace where the primary basis mismatch occurs, applies the rotation, and projects it back. This strategy effectively reduces the computational complexity of residual alignment from $\mathcal{O}(D^2)$ to $\mathcal{O}(D)$, ensuring high efficiency without compromising performance.

To evaluate the effectiveness of ReSpinQuant, we conducted extensive experiments across the Llama-2 (Touvron et al., 2023), Llama-3, and 3.2 families (Grattafiori et al., 2024), achieving state-of-the-art performance in W4A4 and W3A3. ReSpinQuant leverages $63\times$ more learnable parameters (1091.0M) than SpinQuant (17.3M) to maximize representational power. Crucially, through offline weight fusion, the effective online parameter footprint is reduced to just 8.4M—approximately $0.1\%$ of an 8B model. It also incurs a negligible computational overhead of merely $\sim 0.2\%$ (32.3M vs. 15.37T MACs).

Our contributions are summarized as follows:

- **ReSpinQuant Framework:** We introduce a framework that maximizes representational power by employing full-size layer-wise rotations, which are fully fused into weight matrices offline. Combined with subspace residual rotations, our approach achieves high expressivity without compromising efficiency.

- **Empirical Analysis:** We demonstrate that rotation matrices optimized with the Cayley optimizer exhibit strong proximity to their initialization, implying that layer-wise discrepancies are low-rank in nature.

- **Efficiency:** We validate the computational efficiency of our design. We demonstrate that restricting the online overhead to a low-rank subspace incurs negligible computational cost, making ReSpinQuant a practical solution for real-world deployment.

- **SOTA Performance:** We validate ReSpinQuant on LLaMA-2 and LLaMA-3 models. In W4A4 and the challenging W3A3 settings, our method outperforms existing baselines in both perplexity and zero-shot accuracy benchmarks.

## 2. Related Work

### 2.1. Post-Training Quantization and Activation Outliers

Post-Training Quantization (PTQ) has become an essential technique for efficient LLM deployment. While weight-only quantization methods (Frantar et al., 2023; Lin et al., 2024) have achieved success, quantizing activations remains challenging due to the presence of systematic outliers whose activation channels show magnitudes significantly larger than the others (Dettmers et al., 2022). These outliers stretch the dynamic range, causing severe quantization noise in low-bit settings (*e.g.*, W4A4).

Early approaches like SmoothQuant (Xiao et al., 2023) addressed this by migrating the quantization difficulty from activations to weights via channel-wise scaling. However, scaling alone is geometrically restricted to axis-aligned transformations and often fails to fully suppress outliers in lower bit-widths (*e.g.*, under 4-bit).

### 2.2. Rotation-based Quantization

To overcome the limitations of scaling, rotation-based methods introduce orthogonal transformations to rotate the activation space, effectively redistributing outliers across all dimensions. QuaRot (Ashkboos et al., 2024) utilizes fixed randomized Hadamard matrices to "flatten" the activation distribution. Building on this, SpinQuant (Liu et al., 2025) employs learnable rotation matrices optimized via the Cayley optimizer to further minimize quantization error.

A critical design choice in these methods is the scope of the rotation. SpinQuant enforces a **globally shared rotation** across the model. This global constraint ensures that the input and output of attention and FFN blocks share the same basis, allowing the activation rotation to be fused into the weights of the preceding layer offline, without requiring inverse transformations during inference. However, this "one-size-fits-all" approach prevents the model from adapting to the distinct outlier characteristics of individual layers, creating a bottleneck in achievable accuracy.

## 2.3. Layer-Wise Transformation for Quantization

Recent studies have sought to relax the global constraint to improve performance. OSTQuant (Hu et al., 2025) introduces layer-wise parameters but restricts them to scaling vectors to avoid computational overhead. While efficient, its expressivity is limited compared to full rotations.

FlatQuant (Sun et al., 2025) takes a different approach by applying distinct affine transformations for each layer, achieving state-of-the-art accuracy by tailoring the basis to local outlier patterns. However, this layer-wise flexibility prevents the offline fusion of activation transformations into the preceding layer's weights. Unlike SpinQuant, FlatQuant requires explicit activation transformations that cannot be pre-merged. Although FlatQuant mitigates the parameter count using Kronecker products, the alignment still requires an $\mathcal{O}(D^{1.5})$ overhead per token. Without the support of dedicated hardware kernels, this additional matrix multiplication significantly degrades throughput.

Our work bridges this gap by enabling the activation transformations of layer-wise methods to be merged into weights, as in global rotation methods. This fusion allows ReSpinQuant to maintain fully dense, layer-wise learnable rotations to maximize expressivity without losing efficiency. ReSpinQuant architecture also applies rotation to the residual connection of each layer, and approximates this residual transition as a low-rank subspace rotation, achieving $\mathcal{O}(D)$ efficiency without sacrificing the benefits of dense rotations.

## 3. Method

In this section, we present **ReSpinQuant**, a framework designed to resolve the computational inefficiency of layer-wise rotation. We first establish the preliminaries of rotation-based quantization. We then introduce our architecture, which enables weight matrices to fully absorb layer-wise activation rotations, and demonstrate how we resolve the basis mismatch problem via subspace residual rotation.

### 3.1. Preliminaries: Rotation-based Quantization

**Rotation for Outlier Suppression.**   Consider a standard linear layer in a Transformer, computing $\mathbf{Y} = \mathbf{X}\mathbf{W}^\top$, where $\mathbf{X} \in \mathbb{R}^{T \times D}$ is the activation and $\mathbf{W} \in \mathbb{R}^{D_o \times D}$ is the weight matrix for a token length $T$ and input/output dimension $D/D_o$. Direct quantization of $\mathbf{X}$ often incurs significant error due to activation outliers.

Rotation-based quantization methods (Ashkboos et al., 2024; Liu et al., 2025) mitigate this by introducing an orthogonal rotation matrix $\mathbf{R} \in \mathbb{R}^{D \times D}$ (where $\mathbf{R}\mathbf{R}^\top = \mathbf{I}$). By rotating the channels, it spreads outliers uniformly across all dimensions. Exploiting the property of orthogonal trans-

formations, the linear operation can be rewritten as:

$$\mathbf{Y} = \mathbf{X}\mathbf{W}^\top = \mathbf{X}(\mathbf{R}\mathbf{R}^\top)\mathbf{W}^\top = (\mathbf{X}\mathbf{R})(\mathbf{W}\mathbf{R})^\top. \quad (1)$$

Let $\tilde{\mathbf{X}} = \mathbf{X}\mathbf{R}$ and $\tilde{\mathbf{W}} = \mathbf{W}\mathbf{R}$ denote the rotated activations and weights, respectively. Since $\mathbf{R}$ is mathematically fused into the weights offline ($\tilde{\mathbf{W}}$), no additional computation is required during inference.

The quantization is then performed on the rotated domains:

$$\mathbf{Y} \approx Q(\mathbf{X}\mathbf{R})Q(\mathbf{W}\mathbf{R})^\top = Q(\tilde{\mathbf{X}})Q(\tilde{\mathbf{W}})^\top. \quad (2)$$

The goal is to learn an optimal $\mathbf{R}$ such that the rotated activation $\tilde{\mathbf{X}}$ exhibits a reduced dynamic range with fewer outliers, thereby minimizing the quantization error $\|\mathbf{Y} - Q(\tilde{\mathbf{X}})Q(\tilde{\mathbf{W}})^\top\|_F^2$. To strictly enforce the orthogonality of $\mathbf{R}$ during training, we employ the Cayley optimizer (Li et al., 2020), which performs optimization directly on the orthogonal manifold.

While the rotation for weights ($\tilde{\mathbf{W}} = \mathbf{W}\mathbf{R}$) is easily fused offline, handling the activation rotation ($\tilde{\mathbf{X}} = \mathbf{X}\mathbf{R}$) poses a significant implementation challenge. Global rotation methods resolve this by applying a shared rotation, allowing the activation transformation to be pre-multiplied into the output weights of the previous layer, resulting in negligible online overhead.

Conversely, layer-wise methods utilize distinct rotations for each layer, causing a basis mismatch. This mismatch prevents fusion into the preceding layer, necessitating online activation computation. Consequently, prior work was forced to compromise expressivity by using structured transformations instead of full-size matrices. Our ReSpinQuant overcomes this limitation: we utilize full-size rotation matrices by fusing them into the weights of the preceding layer, and effectively resolve the basis mismatch problem via subspace residual rotation approximation.

### 3.2. Overall Architecture

Figure 2 illustrates the comprehensive architecture of ReSpinQuant applied to a standard Transformer layer, comprising Multi-Head Self-Attention (MHSA) and Feed-Forward Network (FFN) blocks.

**Layer-wise Rotation Assignment.**   To maximize the outlier suppression capability, we assign distinct learnable orthogonal matrices to different stages of the network. Let $L$ denote the total number of layers. For the $i$-th layer:

- $\mathbf{R}_1^i$: Rotates the input activation of the MHSA block and output activation of FFN block.

- $\mathbf{R}_2^i$: Rotates the input of FFN block and output of the MHSA block.

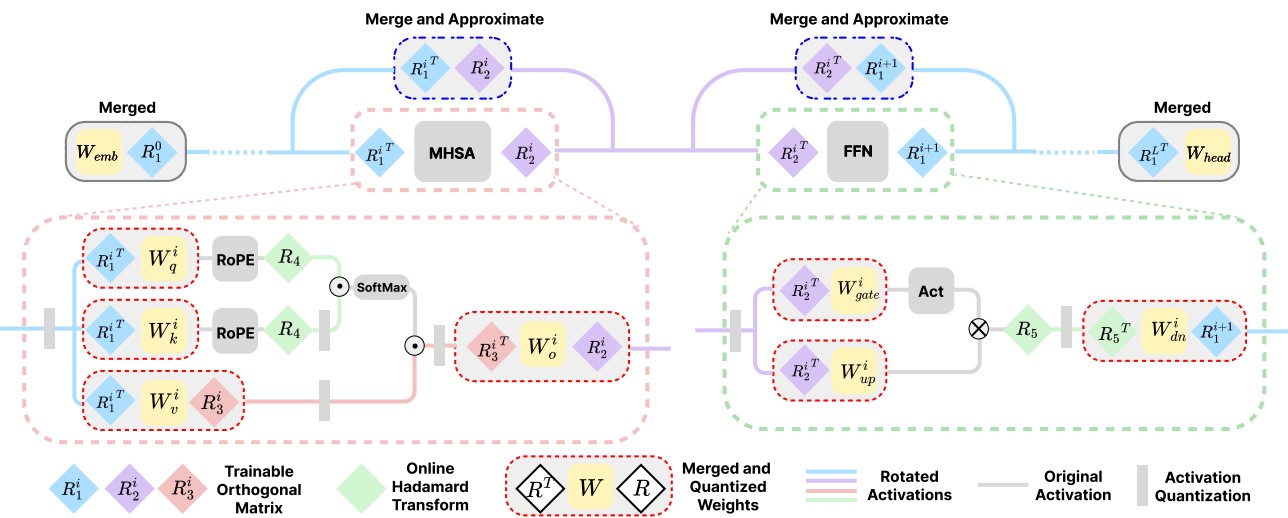

*Figure 2.* **Overview of the ReSpinQuant architecture.** Unlike global rotation methods that enforce a shared basis, ReSpinQuant assigns distinct learnable orthogonal matrices ($\mathbf{R}_1^i$, $\mathbf{R}_2^i$, etc.) to each layer $i$ to maximize outlier suppression. **(1) Offline Weight Fusion:** To ensure offline weight integration, rotation matrices are mathematically absorbed into the weight matrices at inference phase (dashed red boxes, "Merged and Quantized Weights"), leaving the main computation unchanged. **(2) Subspace Residual Approximation:** The resulting basis mismatch in residual connections is resolved using our proposed *Subspace Residual Rotation Approximation* (dash-dot blue boxes, "Merge and Approximate"). This replaces the expensive dense basis transition with a low-rank correction, reducing alignment complexity from $\mathcal{O}(D^2)$ to $\mathcal{O}(D)$ while maintaining high model expressivity.

- $\mathbf{R}_3^i$: An intermediate rotation applied within the attention mechanism (*e.g.*, Value projection) to further decorrelate internal features.

- $\mathbf{R}_4, \mathbf{R}_5$: Structured rotations implemented via the Fast Hadamard Transform. Following the SpinQuant (Liu et al., 2025) protocol, these are computed online to handle specific activation flows. They incur a computational complexity of $\mathcal{O}(D \log D)$.

This configuration allows the model to adapt the basis to the outlier patterns of each transformer block (MHSA, FFN).

**Offline Weight Merging.** As shown in the "Merged and Quantized Weights" blocks in Figure 2, linear transformations within MHSA or FFN absorb the rotation matrices. For instance, the value projection weight $\mathbf{W}_v$ is fused as $\tilde{\mathbf{W}}_v = \mathbf{R}_1^{i\top} \mathbf{W}_v \mathbf{R}_3$, and the output projection $\mathbf{W}_o$ absorbs the transition from the internal state to the output basis $\tilde{\mathbf{W}}_o = \mathbf{R}_3^{i\top} \mathbf{W}_o \mathbf{R}_2$. Consequently, the main computation path remains mathematically equivalent to standard linear layers, incurring low inference overhead.

**The Basis Mismatch Problem.** While the above formulation holds for transformer blocks, a challenge arises in residual connections when applying layer-wise rotations. Let $\mathbf{R}_{in}$ and $\mathbf{R}_{out}$ denote the optimal layer-wise rotation matrices for the input and output of each MHSA or FFN block. Then the residual connection $x_{out} = x_{in} + \text{Block}(x_{in})$ in

the rotated coordinate system is expressed as:

$$\tilde{x}_{out} = \underbrace{\mathbf{R}_{out}\mathbf{R}_{in}^T}_{\mathbf{T}} \tilde{x}_{in} + \underbrace{\mathbf{R}_{out}\text{Block}(\mathbf{R}_{in}^T \tilde{x}_{in})}_{\text{Merged}} \quad (3)$$

In methods employing a global rotation strategy (*e.g.*, SpinQuant), setting $\mathbf{R}_{in} = \mathbf{R}_{out} = \mathbf{R}$ results in $\mathbf{T} = \mathbf{I}$, thereby eliminating residual computational overhead. However, this restricts the model's expressivity by forcing a single rotation basis across all layers. To address this, we employ full-size layer-wise rotation matrices to maximize expressivity, while simultaneously minimizing overhead by approximating the residual rotation $\mathbf{T}$ via a subspace rotation approximation, reducing the complexity from $\mathcal{O}(D^2)$ to $\mathcal{O}(D)$.

### 3.3. Subspace Residual Rotation Approximation

**Empirical Observation.** Our method relies on a key empirical finding regarding the optimization trajectory of rotation matrices. Following SpinQuant (Liu et al., 2025), we initialize rotations with a Hadamard matrix $\mathbf{H}$ and optimize them via Cayley optimizer.

As shown in Figure 3, and Figure 4, the learned rotation matrices ($\mathbf{R}_1, \mathbf{R}_2$) do not deviate significantly from the initial Hadamard structure after convergence. Consequently, the residual rotation matrix $\mathbf{T}$ exhibits strong diagonal dominance:

$$\mathbf{T} = \mathbf{R}_{out}\mathbf{R}_{in}^T \approx \mathbf{H}\mathbf{H}^T = \mathbf{I} \quad (4)$$

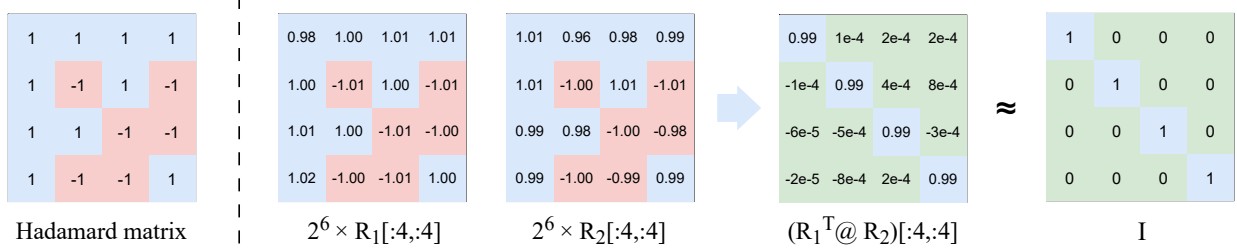

*Figure 3.* **Visualization of the trained rotation matrices optimized by Cayley optimizer.** We display the sub-blocks [:4, :4] of two rotation matrices ($\mathbf{R}_1, \mathbf{R}_2 \in \mathbb{R}^{4096 \times 4096}$) of Llama-3 8B layer 16, and their relative transformation ($\mathbf{R}_1^\top \mathbf{R}_2$) after training. Despite optimization, the matrices deviate minimally from the Hadamard initialization. Crucially, the relative transformation $\mathbf{R}_1^\top \mathbf{R}_2$ remains sparse and diagonally dominant, motivating the use of our residual subspace approximation.

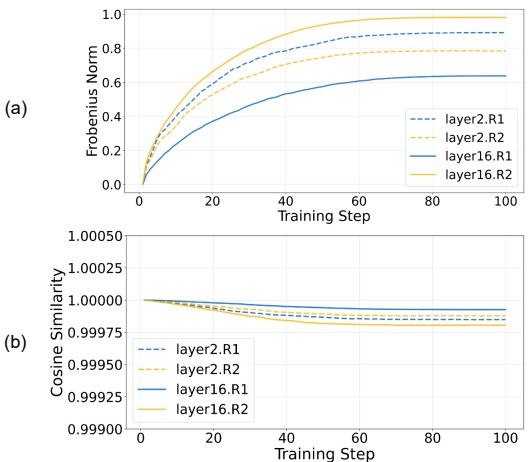

*Figure 4.* **Optimization dynamics of rotation matrices on LLaMA-3 8B (Layers 2, 16). (a)** The Frobenius norm of the deviation from the initialization, calculated as $\|\mathbf{R} - \mathbf{R}_{\text{init}}\|_F$. The curve indicates stable convergence during training. **(b)** The cosine similarity between the optimized rotation $\mathbf{R}$ and the initialization $\mathbf{R}_{\text{init}}$ remains consistently high. Together, these plots confirm that while the learned rotations successfully converge to an optimal state, they do not deviate significantly from their initial values.

**Principal Subspace Identification.** We define the deviation from the identity as $\Delta\mathbf{T} = \mathbf{T} - \mathbf{I}$. We perform Singular Value Decomposition (SVD) on this deviation to identify the principal directions of basis mismatch:

$$\mathbf{Q}, \mathbf{S}, \mathbf{V}^T = \text{SVD}(\mathbf{T} - \mathbf{I}) \tag{5}$$

We truncate the decomposition to keep only the top-$r$ singular vectors, forming a projection matrix $\mathbf{Q} \in \mathbb{R}^{D \times r}$, where $r \ll D$.

**Orthogonal Subspace Projection.** Once the subspace basis $\mathbf{Q}$ is identified, we derive the optimal rotation $\hat{\mathbf{R}}_{\text{sub}} \in \mathbb{R}^{r \times r}$ within this subspace. We first project the full transition matrix $\mathbf{T}$ onto the subspace:

$$\mathbf{T}_{\text{sub}} = \mathbf{Q}^T \mathbf{T} \mathbf{Q} \quad \in \mathbb{R}^{r \times r} \tag{6}$$

Since the projection does not strictly preserve orthogonality, we extract the closest orthogonal matrix using the polar decomposition. Specifically, we perform SVD on the projected component:

$$\mathbf{U}_{sub}, \boldsymbol{\Sigma}_{sub}, \mathbf{V}_{sub}^T = \text{SVD}(\mathbf{T}_{\text{sub}}) \tag{7}$$

The orthogonalized subspace rotation is obtained as $\hat{\mathbf{R}}_{\text{sub}} = \mathbf{U}_{sub}\mathbf{V}_{sub}^T$, which ensures $\hat{\mathbf{R}}_{\text{sub}} \in SO(r)$.

**Rotation Approximation.** We approximate the full rotation $\mathbf{T}$ by applying the transformation only within the identified subspace, while leaving the orthogonal complement space invariant. The approximated transition $\hat{\mathbf{T}}$ is formulated as:

$$\hat{\mathbf{T}} = \underbrace{(\mathbf{I} - \mathbf{Q}\mathbf{Q}^T)}_{(D-r) \text{ identity mapping}} + \underbrace{\mathbf{Q}\hat{\mathbf{R}}_{\text{sub}}\mathbf{Q}^T}_{\text{subspace rotation}} \tag{8}$$

$$= \mathbf{I} + \mathbf{Q}(\hat{\mathbf{R}}_{\text{sub}} - \mathbf{I}_r)\mathbf{Q}^T$$

where $\mathbf{I} - \mathbf{Q}\mathbf{Q}^T$ represents the projection onto the complement space. This approximation allows us to effectively estimate $\mathbf{T}$ while strictly ensuring that the resulting matrix $\hat{\mathbf{T}}$ remains within the Special Orthogonal group $SO(D)$.

In practice, we implement this efficiently without constructing the $D \times D$ matrix $\hat{\mathbf{T}}$. The operation on the input vector $\tilde{x}_{in}$ is performed as follows (see Figure 5):

1. **Projection:** Project the input into the low-rank subspace:

$$y = \mathbf{Q}^T \tilde{x}_{in} \quad \in \mathbb{R}^r \tag{9}$$

2. **Subspace Transformation:** Apply the learnable dense transform within the $r$-dimensional space. We define the effective subspace matrix $\mathbf{M} = \hat{\mathbf{R}}_{\text{sub}} - \mathbf{I}_r$ to merge the add operations:

$$z = \mathbf{M}y \quad \in \mathbb{R}^r \tag{10}$$

3. **Re-projection & Residual Add:** Project back to the original dimension and add to the input:

$$\tilde{x}_{out} = \tilde{x}_{in} + \mathbf{Q}z \tag{11}$$

*Figure 5.* **Illustration of the subspace residual rotation approximation.** We approximate the dense rotation matrix ($\mathbf{T} \in \mathbb{R}^{D \times D}$) that aligns the residual bases into a low-dimensional subspace to minimize the computational overhead. During inference, the input features are projected into a subspace of rank $r$ via $\mathbf{Q}$, rotated by the approximated matrix ($\hat{\mathbf{R}}_{\text{sub}} \in \mathbb{R}^{r \times r}$), and re-projected via $\mathbf{Q}^\top$. The orthogonal complement passes through the identity path ($\mathbf{I} - \mathbf{Q}\mathbf{Q}^\top$). The alignment complexity is reduced from quadratic $\mathcal{O}(D^2)$ to linear $\mathcal{O}(D)$.

This structure ensures that the residual rotation computation is confined to $r$ dimensions, significantly reducing overhead.

## 4. Experiments

### 4.1. Experimental Setup

**Models and Datasets.** We evaluated **ReSpinQuant** on the Llama 2 (7B, 13B) (Touvron et al., 2023), Llama 3 (8B), and Llama 3.2 (1B, 3B) (Grattafiori et al., 2024) families. Following standard protocols, we use the WikiText-2 dataset (Merity et al., 2017) for calibration and perplexity (PPL) evaluation. For zero-shot reasoning capabilities, we report the accuracy on nine diverse downstream tasks: BoolQ (Clark et al., 2019), PIQA (Bisk et al., 2020), SIQA (Sap et al., 2019), HellaSwag (Zellers et al., 2019), WinoGrande (Sakaguchi et al., 2021), ARC-Easy and ARC-Challenge (Clark et al., 2018), OBQA (Mihaylov et al., 2018), and LAMBADA (Paperno et al., 2016).

**Baselines and Implementation Details.** We compare our method against state-of-the-art PTQ baselines:

- **Basic PTQ:** RTN, GPTQ (Frantar et al., 2023).

- **Rotation-based (Global rotation):** QuaRot (Ashkboos et al., 2024) (random Hadamard matrix), SpinQuant (Liu et al., 2025) (our primary baseline).

- **Layer-wise Approaches:** OSTQuant (Hu et al., 2025) (layer-wise scaling) and FlatQuant (Sun et al., 2025) (affine transform).

Consistent with SpinQuant, we optimize the rotation matrices using the Cayley optimizer (Li et al., 2020) with a calibration set of 800 random segments from WikiText-2. For ReSpinQuant, we optimize full-size layer-wise rotations and fuse them into the weights of Attention and FFN blocks. During inference, we employ a rank $r = 32$ approximation for the residual transition matrix as the default configuration, a choice derived from our ablation studies (See Table 5). To ensure a fair comparison, all evaluated methods (QuaRot, SpinQuant, OSTQuant, FlatQuant, and ReSpinQuant) employ GPTQ for weight quantization following transformation optimization, utilizing a fixed weight clipping strategy. Unlike OSTQuant and FlatQuant, which introduce both structural modifications and require specialized objective functions (*e.g.*, KL divergence or layer-wise losses), ReSpinQuant is optimized using only the standard cross-entropy loss on the calibration dataset, identical to the SpinQuant protocol. Additional implementation details are provided in Appendix A. All experiments were conducted on an NVIDIA H100 GPU.

### 4.2. Main Results: W4A4 and W3A3 Quantization

Table 1 presents the comparative results of ReSpinQuant against state-of-the-art baselines across LLaMA-2, LLaMA-3, and LLaMA-3.2 families.

**Performance on W4A4 Quantization.** In the standard W4A4 setting (weights, activations, and key-value caches are quantized to 4-bit), ReSpinQuant consistently achieves the lowest perplexity (PPL) and highest zero-shot accuracy across all evaluated models. While global rotation methods like QuaRot and SpinQuant already provide strong baselines compared to RTN or GPTQ, ReSpinQuant further pushes the performance boundary. For instance, on LLaMA-3 8B, ReSpinQuant reduces the PPL to 7.24, outperforming QuaRot (7.82) and SpinQuant (7.50). This demonstrates that the enhanced expressivity from full layer-wise rotations effectively captures the outliers that a single global rotation cannot address, without compromising inference efficiency.

Notably, our results underscore the critical necessity of quantization by demonstrating that a quantized larger model offers a superior trade-off compared to a full-precision smaller model. As shown in Table 1, LLaMA-3.2 3B quantized with ReSpinQuant (W4A4) achieves a PPL of 9.06 and 58.84% accuracy, surpassing the full-precision LLaMA-3.2 1B (PPL 9.76, 55.75%). Considering that a 4-bit 3B model requires less memory than a 16-bit 1B model, this confirms that high-quality quantization effectively shifts the Pareto frontier, allowing deployment of superior performance within a more constrained computational and memory budget.

**Performance on W3A3 Quantization.** The advantage of ReSpinQuant becomes significantly more pronounced in the challenging W3A3 regime. As shown in Table 1, existing

*Table 1.* **Main results on LLaMA-2, LLaMA-3, and LLaMA-3.2 families.** We report Perplexity (PPL) on WikiText-2 and Zero-shot Accuracy (Avg.). **ReSpinQuant** consistently outperforms the global rotation baseline (SpinQuant) and matches or exceeds the layer-wise baseline across diverse model scales.

| Bits | Method | LLaMA-2 7B | | LLaMA-2 13B | | LLaMA-3 8B | | LLaMA-3.2 1B | | LLaMA-3.2 3B | |
|---|---|---|---|---|---|---|---|---|---|---|---|
| | | PPL ↓ | 0-shot[9] ↑ | PPL ↓ | 0-shot[9] ↑ | PPL ↓ | 0-shot[9] ↑ | PPL ↓ | 0-shot[9] ↑ | PPL ↓ | 0-shot[9] ↑ |
| 16-16 | FP16 | 5.47 | 65.08 | 4.88 | 67.77 | 6.13 | 68.05 | 9.76 | 55.75 | 7.81 | 63.44 |
| **W4A4** | RTN | 1753.70 | 33.69 | 3575.44 | 31.24 | 219.82 | 36.74 | 330.79 | 33.52 | 266.80 | 35.15 |
| | GPTQ | 9787.38 | 34.89 | 2841.03 | 31.46 | 187.49 | 35.74 | 223.08 | 33.81 | 168.41 | 36.51 |
| | QuaRot | 6.12 | 62.03 | 5.37 | 65.25 | 7.82 | 62.90 | 14.59 | 46.81 | 9.99 | 57.22 |
| | SpinQuant | 5.98 | 61.60 | 5.30 | 65.58 | 7.50 | 64.53 | 13.52 | 48.35 | 9.46 | 56.78 |
| | OSTQuant | 5.95 | 62.32 | 5.28 | 65.41 | 7.42 | 64.35 | 13.23 | 48.80 | 9.27 | 58.67 |
| | FlatQuant | 6.08 | 61.97 | 5.32 | 65.04 | 7.73 | 62.72 | 13.63 | 48.20 | 9.57 | 58.06 |
| | **ReSpinQuant** | **5.74** | **62.49** | **5.13** | **65.82** | **7.24** | **64.65** | **13.03** | **49.57** | **9.06** | **58.84** |
| **W3A3** | RTN | 19519 | 32.56 | 14960 | 32.59 | 77055 | 32.71 | 115358 | 31.62 | 22132 | 31.28 |
| | GPTQ | 99691 | 33.78 | 6837 | 31.84 | 44824 | 32.31 | 56762 | 31.13 | 10753 | 31.48 |
| | QuaRot | 19.26 | 40.94 | 12.78 | 43.67 | 98.04 | 33.95 | 812.46 | 32.07 | 172.33 | 32.94 |
| | SpinQuant | 9.69 | 49.15 | 7.36 | 55.10 | 15.07 | 48.77 | 69.70 | 35.02 | 21.05 | 42.16 |
| | OSTQuant | 9.53 | 48.44 | 7.10 | 56.36 | 14.29 | 47.34 | 59.71 | 35.71 | 21.27 | 41.16 |
| | FlatQuant | 13.32 | 45.43 | 9.27 | 50.95 | 133.52 | 33.81 | 543.66 | 32.41 | 84.47 | 34.13 |
| | **ReSpinQuant** | **8.57** | **50.14** | **6.91** | **56.58** | **13.09** | **50.74** | **49.90** | **36.65** | **18.61** | **43.63** |

*Table 2.* **Comparison of learnable parameters (training vs. online) and multiply-accumulate operations (MACs)** for rotation-based quantization methods on Llama3-8B model. ReSpinQuant maximizes learnable parameters during training ($\mathcal{O}(L \cdot D^2)$) for high expressivity, but most are fused into weights, leaving only low-rank subspace residual rotation parameters for online inference.

| Method | Transform Scope | Learnable Parameters | | | Additional Computation | |
|---|---|---|---|---|---|---|
| | | Parameter Space | Trainable Params | Online Params | Complexity | MACs |
| SpinQuant (Liu et al., 2025) | Global | $\mathcal{O}(D^2)$ | 17.3M | − | $\mathcal{O}(D \log D)$ | 15.4M |
| OSTQuant (Hu et al., 2025) | Layer-wise | $\mathcal{O}(D^2 + L \cdot D)$ | 21.7M | − | $\mathcal{O}(D \log D)$ | 15.4M |
| FlatQuant (Sun et al., 2025) | Layer-wise | $\mathcal{O}(L \cdot D)$ | 5.8M | 5.8M | $\mathcal{O}(D^{1.5})$ | 198.1M |
| **ReSpinQuant (Ours)** | **Layer-wise** | $\mathcal{O}(L \cdot D^2)$ | **1091.0M** | **8.4M** | $\mathcal{O}(D \log D + rD)$ | **32.3M** |

methods struggle to maintain stability as bit-width decreases. Compared to SpinQuant, ReSpinQuant achieves substantial gains. On the LLaMA-3.2 1B model, ReSpinQuant lowers the PPL by over 19.8 points ($69.70 \rightarrow 49.90$). These results confirm that our method preserves the representational power of layer-wise rotations, enabling ReSpinQuant to operate reliably even under extreme quantization constraints where other methods falter.

## 4.3. Efficiency Analysis

We provide a comprehensive analysis of the computational efficiency of ReSpinQuant, demonstrating how our "Train-Large, Infer-Small" strategy translates into real-world performance.

**Parameter Efficiency and Computational Complexity.** Table 2 contrasts the learnable parameters during training versus inference. A distinctive feature of ReSpinQuant is its ability to disentangle training expressivity from inference cost. During the optimization phase, ReSpinQuant utilizes a massive parameter space of $\mathcal{O}(L \cdot D^2)$ to find precise ro-

tation parameters. However, we compress these parameters into a low-rank subspace ($r = 32$) for online inference. This results in an additional computational cost of only 32.3M MACs, representing a negligible 0.2% of the original cost (15.37T MACs).

**Inference Latency.** To validate the real-world impact of our theoretical efficiency, we measured the end-to-end latency on an NVIDIA H100 GPU (Table 3). ReSpinQuant demonstrates inference speeds comparable to the SpinQuant baseline, which uses a globally merged rotation (low overhead). For instance, at a batch size of 16, the Time To Iterative-Token (TTIT) latency increases marginally from 160.95 ms (SpinQuant) to 163.81 ms (ReSpinQuant), representing an overhead of $\sim 1.7\%$. It is worth noting that these measurements were obtained without specialized hardware optimizations for 4-bit arithmetic. We anticipate that incorporating dedicated low-level kernels for W4A4 operations would further reduce latency, unlocking even greater inference throughput.

*Table 3.* **End-to-end latency comparison on LLaMA-3 8B (W4A4).** Measured on an NVIDIA H100 GPU with a sequence length of $T = 2048$. We report Time to First Token (TTFT) which measures the prefill latency, and Time to Iterative Token (TTIT) representing the decoding latency. ReSpinQuant (rank $r = 32$) demonstrates comparable inference speed to the globally-rotated SpinQuant baseline across all batch sizes.

| Batch Size | SpinQuant | | ReSpinQuant | |
|---|---|---|---|---|
| | TTFT (ms) | TTIT (ms) | TTFT (ms) | TTIT (ms) |
| 1 | 389.20 | 120.38 | 394.75 | 123.45 |
| 2 | 757.18 | 122.29 | 766.03 | 126.90 |
| 4 | 1,453.42 | 126.42 | 1,473.54 | 129.53 |
| 8 | 2,774.59 | 141.23 | 2,886.14 | 144.80 |
| 16 | 5,589.04 | 160.95 | 5,661.14 | 163.81 |

*Table 4.* **Comparison of calibration overhead** measured on a single NVIDIA H100 GPU. Although a larger parameter space leads to increased training time, ReSpinQuant still has a practical calibration time, finishing within an hour for most models.

| Method | Training Time (Minute) | | |
|---|---|---|---|
| | LLaMA-2 7B | LLaMA-2 13B | LLaMA-3 8B |
| SpinQuant | 17m | 23m | 17m |
| OSTQuant | 15m | 26m | 12m |
| FlatQuant | 45m | 74m | 45m |
| **ReSpinQuant** | 43m | 61m | 42m |

**Training Overhead.** Finally, we examine the training overhead associated with the expanded parameter space. As shown in Table 4, optimizing layer-wise rotations in ReSpinQuant inevitably leads to an increase in calibration time compared to the global rotation baseline (SpinQuant). For LLaMA-3 8B, the training time increases from 17 minutes to 42 minutes. However, this overhead remains manageable, completing within an hour on a single NVIDIA H100 GPU. Given the substantial performance gains, we consider this moderate increase in offline training cost to be a worthwhile trade-off.

### 4.4. Ablation Studies

**Impact of Approximation Rank $r$.** We investigate the impact of the approximation rank $r$ on quantization performance. Table 5 summarizes the perplexity (PPL) and zero-shot accuracy on LLaMA-3 8B under the W3A3 quantization setting.

As shown in the results, even a minimal subspace rank yields substantial improvements. Increasing $r$ from 0 (Identity, no residual rotation) to 8 drastically reduces PPL from 20.03 to 14.20, indicating that the principal directions of the basis mismatch are concentrated in a very small subspace.

We observe that performance continues to improve as the rank increases to 64 and 128. However, considering the bal-

*Table 5.* **Ablation study on approximation rank $r$ using LLaMA-3-8B (W3A3).** Performance consistently improves as the rank increases. We select $r = 32$ as the default setting, as it achieves a favorable trade-off between accuracy and parameter efficiency.

| Rank $r$ | PPL $\downarrow$ | Zero-shot Avg. $\uparrow$ |
|---|---|---|
| 0 (Identity) | 20.03 | 46.94 |
| 8 | 14.20 | 49.77 |
| 16 | 13.59 | 50.14 |
| **32** | 13.09 | 50.74 |
| 64 | 12.84 | 51.29 |
| 128 | 12.80 | 50.80 |
| 4096 (Full-Rank) | 12.52 | 51.22 |

ance between model accuracy and computational efficiency, we identify $r = 32$ as the optimal configuration. Notably, our chosen default rank of $r = 32$ achieves a PPL of 13.09, which is comparable to the $r = 128$ performance (PPL 12.80). Therefore, we select $r = 32$ to maximize efficiency while retaining the majority of the accuracy gains.

## 5. Conclusion

In this paper, we present ReSpinQuant, a novel rotation-based post-training quantization framework that effectively achieves superior expressivity with negligible overhead. This is achieved by fully fusing layer-wise rotations into weight matrices, while simultaneously handling the residual with a low-rank correction. This enables ReSpinQuant itself to find an optimal balance between global rotation methods and layer-wise transformation methods: it achieves high efficiency by following the weight merging strategy of global rotation methods, and attaining high capability by leveraging layer-wise matrices as in other layer-wise transformation methods. Extensive experiments on the LLaMA-2 and LLaMA-3 families demonstrate that our method achieves state-of-the-art performance in W4A4 and W3A3 settings, matching the accuracy of computationally intensive layer-wise baselines while maintaining the high inference throughput of global rotation methods.

## 6. Limitation

Our study focuses exclusively on structural innovations under a standard global calibration objective. Unlike prior works that leverage specialized layer-wise loss functions or training schemes, we did not explore these avenues; thus, integrating advanced optimization objectives could yield further improvements. Additionally, computational constraints (a single GPU) limited our evaluation to models up to 13B parameters, precluding tests on larger-scale models. Finally, while we validated theoretical efficiency, the development of a dedicated hardware-efficient kernel implementation remains for future work.

## Impact Statement

This paper presents work whose goal is to advance the field of efficient Machine Learning. Our proposed method, ReSpinQuant, significantly reduces the computational and memory barriers for deploying Large Language Models (LLMs). By enabling high-quality inference (*e.g.*, W4A4, W3A3) on consumer-grade hardware or edge devices, this work contributes to the democratization of AI, allowing researchers and practitioners with limited computational resources to utilize state-of-the-art models.

Furthermore, by reducing the inference overhead and memory bandwidth requirements, our approach aligns with the goals of Green AI, potentially lowering the carbon footprint associated with large-scale model deployment. However, we acknowledge that lowering the barrier to entry for powerful LLMs also carries the risk of facilitating malicious uses, such as the generation of misinformation or harmful content on local devices without API-level safety moderation. We believe that the development of efficient quantization must go hand-in-hand with research into robust model safety and alignment.

## Acknowledgements

This work was supported by the Korean Government through the grants from IITP (RS-2021-II211343, RS-2022-II220320, RS-2025-25442338) and Samsung Electronics Co., Ltd(IO251219-14870-01).

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

# A. Additional Implementation Details

In this section, we provide detailed configurations for our experiments to ensure reproducibility.

**General Experimental Setup.** All experiments were conducted on an NVIDIA H100 (80GB) GPU. We implemented our framework using PyTorch and the HuggingFace transformers library. Across all methods, we consistently use WikiText-2 (Merity et al., 2017) as the calibration dataset. To evaluate the zero-shot performance, we utilized the `lm-evaluation-harness` library (version `0.4.4`).

**RTN / GPTQ (Frantar et al., 2023).** Since GPTQ is applicable only to weight quantization, any reference to "GPTQ quantization" in our experiments implies applying GPTQ to weights while using Round-to-Nearest (RTN) for activations. Conversely, "RTN quantization" refers to applying RTN to both weights and activations.

**SpinQuant (Liu et al., 2025).** We follow the official default configuration of SpinQuant. Specifically, we apply asymmetric quantization to activations and Key-Value (KV) caches, while employing symmetric quantization for weights. The optimization process is conducted for 100 steps with a learning rate of 1.5, utilizing a cosine learning rate scheduler. For weight quantization, we employ GPTQ combined with fixed weight clipping.

**OSTQuant (Hu et al., 2025).** For OSTQuant, we noted discrepancies between the hyperparameters reported in the original paper and those provided in the official implementation. As the specific model-wise hyperparameters used in the paper are not fully disclosed, we adopted the default settings from the latest version of the official GitHub repository (commit hash `ab64362`). Similar to SpinQuant, we use asymmetric quantization for activations and KV caches, and symmetric quantization for weights. Weights are quantized using GPTQ with fixed weight clipping. We set the learning rate for rotation matrices to approximately 0.01689... and for diagonal scaling to 0.001789..., utilizing a linear learning rate scheduler.

**FlatQuant (Sun et al., 2025).** For FlatQuant, we prioritized adhering to the official implementation. Accordingly, we apply asymmetric quantization to KV caches and symmetric quantization to both weights and activations. Since FlatQuant optimizes parameters in a layer-wise manner, we train for 15 epochs with a learning rate of $1 \times 10^{-5}$ using a cosine learning rate scheduler. Although FlatQuant supports both RTN and GPTQ for weight quantization, we utilized GPTQ to ensure a fair comparison with other baselines. Furthermore, while the original FlatQuant paper employs learnable weight and activation clipping, we replaced this with fixed weight clipping in our experiments. This modification was made to align the quantization methodology with other baselines and strictly evaluate the impact of the rotation transformation itself.

**ReSpinQuant (Ours).** We aligned the configuration of ReSpinQuant as closely as possible with SpinQuant to ensure a direct comparison. Consequently, we use asymmetric quantization for activations and KV caches, and symmetric quantization for weights. The optimization is performed for 100 steps. However, a key difference lies in the learning rate; due to the increased number of learnable parameters in our framework, the original learning rate proved insufficient for convergence. We therefore increased the learning rate to 15. We observed that the optimization remains stable even at this high learning rate because the Cayley optimizer constrains the parameters to the orthogonal manifold. Consistent with SpinQuant, we use a cosine learning rate scheduler, employ GPTQ for weight quantization, and apply fixed weight clipping.

# B. Additional Experiment Results

## B.1. Additional Rank Analysis

Table 5 in the main text reports the end-task performance of ReSpinQuant with different approximation ranks. Here, we provide a complementary spectral analysis of the residual transition deviation $\Delta T = T - I$ to further justify why a small subspace rank is sufficient in practice.

Specifically, we measure the spectral concentration of $\Delta T$ on LLaMA-2 7B under W3A3 quantization. Table 6 reports the cumulative singular-value energy captured by the top-$r$ components, as well as the retained Frobenius magnitude of the approximated residual correction. Although $r = 32$ corresponds to only $32/4096 = 0.78\%$ of the full feature dimension, it captures 55.77% of the cumulative energy of $\Delta T$ and retains 73.80% of the Frobenius magnitude of the full residual correction. This confirms that the residual basis mismatch is highly compressible and supports the rank-performance trend observed in the main-text rank ablation.

*Table 6.* Spectral concentration of the residual transition deviation $\Delta T = T - I$ on LLaMA-2 7B under W3A3 quantization.

| Rank $r$ | 0 | 8 | 16 | **32** | 64 | 128 | 4096 |
|---|---|---|---|---|---|---|---|
| Cumulative energy of $\Delta T$ (%) | 0.00 | 51.07 | 53.45 | **55.77** | 58.55 | 62.33 | 100.00 |
| Retained Frobenius magnitude (%) | 0.00 | 70.25 | 72.09 | **73.80** | 75.76 | 78.34 | 100.00 |

## B.2. Additional Bit-Width Results

While the main experiments focus on the aggressive W4A4KV4 and W3A3KV3 settings, we additionally evaluate ReSpin-Quant under broader weight-activation-KV precision settings. Table 7 reports the results for W4A4KV16, W4A8KV8, and W4A8KV16 across five LLaMA-family models.

In the aggressive W3A3KV3, W4A4KV4, W4A4KV16 setting, ReSpinQuant consistently improves over SpinQuant across all evaluated models in both perplexity and zero-shot average accuracy. In the milder W4A8KV8 and W4A8KV16 settings, where SpinQuant is already close to full precision, ReSpinQuant remains comparable to or slightly better than SpinQuant. These results show that the proposed layer-wise rotation and residual subspace correction remain effective across a wider range of bit-width configurations, with larger gains appearing when the quantization budget is more constrained.

*Table 7.* Additional bit-width results on LLaMA-family models. We report PPL and zero-shot accuracy averaged over nine tasks.

| Setting | Method | LLaMA-2 7B | | LLaMA-2 13B | | LLaMA-3 8B | | LLaMA-3.2 1B | | LLaMA-3.2 3B | |
|---|---|---|---|---|---|---|---|---|---|---|---|
| | | PPL $\downarrow$ | 0-shot[9] $\uparrow$ | PPL $\downarrow$ | 0-shot[9] $\uparrow$ | PPL $\downarrow$ | 0-shot[9] $\uparrow$ | PPL $\downarrow$ | 0-shot[9] $\uparrow$ | PPL $\downarrow$ | 0-shot[9] $\uparrow$ |
| FP16 | Full precision | 5.47 | 65.08 | 4.88 | 67.77 | 6.13 | 68.05 | 9.76 | 55.75 | 7.81 | 63.44 |
| W4A4KV16 | SpinQuant | 5.93 | 61.98 | 5.26 | 65.43 | 7.38 | 64.43 | 13.03 | 48.98 | 9.15 | 57.55 |
| | ReSpinQuant | **5.69** | **62.49** | **5.08** | **65.63** | **7.16** | **65.41** | **12.36** | **49.44** | **8.89** | **59.49** |
| W4A8KV8 | SpinQuant | 5.62 | 64.07 | 5.00 | 66.85 | **6.53** | 67.28 | **10.50** | **53.39** | 8.24 | 62.24 |
| | ReSpinQuant | **5.60** | **64.24** | **4.98** | **67.10** | **6.53** | **67.32** | 10.51 | 53.28 | **8.22** | **62.46** |
| W4A8KV16 | SpinQuant | **5.62** | 64.22 | 5.01 | 66.86 | **6.54** | 67.37 | **10.49** | **53.79** | 8.23 | 61.80 |
| | ReSpinQuant | **5.62** | **64.32** | **4.98** | **67.03** | **6.54** | **67.44** | 10.51 | 53.75 | **8.21** | **62.02** |

## B.3. Calibration Dataset Sensitivity

We further evaluate the sensitivity of ReSpinQuant to the calibration dataset by replacing WikiText-2 calibration samples with C4. The results under W4A4KV4 are summarized in Table 8. Across all evaluated model scales, ReSpinQuant consistently improves over SpinQuant in both PPL and zero-shot average accuracy. For example, on LLaMA-2 7B, ReSpinQuant reduces PPL from 8.07 to 7.70 and improves zero-shot average accuracy from 61.67 to 62.40. These results suggest that ReSpinQuant is not highly sensitive to the calibration corpus and maintains robust gains under a different calibration distribution.

## B.4. Detailed zero-shot results

This section presents detailed numerical results corresponding to the experiments discussed in the main text. We report Perplexity (PPL) on WikiText-2 (Merity et al., 2017) and Zero-shot Accuracy across nine benchmarks: BoolQ (Clark et al., 2019), PIQA (Bisk et al., 2020), SIQA (Sap et al., 2019), HellaSwag (HellaS.) (Zellers et al., 2019), WinoGrande (WinoG.) (Sakaguchi et al., 2021), ARC-easy (ARC-e) and ARC-challenge (ARC-c) (Clark et al., 2018), OpenBookQA (OBQA) (Mihaylov et al., 2018), and LAMBADA (LAMB) (Paperno et al., 2016).

Table 9 lists the full W4A4 quantization results for the LLaMA-2 (Touvron et al., 2023), LLaMA-3, and LLaMA-3.2 (Grattafiori et al., 2024) families, including comparisons with baselines. Table 10 provides the experimental results under the W3A3 setting for the same models and tasks.

*Table 8.* Calibration dataset sensitivity under W4A4KV4. We use C4 as the calibration corpus and report PPL and nine-task zero-shot average accuracy.

| Method | LLaMA-2 7B | | LLaMA-2 13B | | LLaMA-3 8B | | LLaMA-3.2 1B | | LLaMA-3.2 3B | |
|---|---|---|---|---|---|---|---|---|---|---|
| | PPL ↓ | 0-shot[9] ↑ | PPL ↓ | 0-shot[9] ↑ | PPL ↓ | 0-shot[9] ↑ | PPL ↓ | 0-shot[9] ↑ | PPL ↓ | 0-shot[9] ↑ |
| SpinQuant | 8.07 | 61.67 | 7.39 | 65.46 | 11.88 | 63.79 | 20.33 | 49.11 | 14.39 | 58.01 |
| ReSpinQuant | **7.70** | **62.40** | **7.29** | **66.06** | **11.50** | **64.67** | **19.82** | **49.48** | **13.99** | **58.64** |

*Table 9.* Detailed W4A4 quantization results across benchmarks.

| Model | Method | BoolQ | PIQA | SIQA | HellaS. | WinoG. | ARC-e | ARC-c | OBQA | LAMB | Avg. (↑) | PPL (↓) |
|---|---|---|---|---|---|---|---|---|---|---|---|---|
| | Full precision | 77.77 | 79.05 | 45.96 | 76.00 | 68.51 | 74.54 | 46.16 | 44.00 | 73.70 | 65.08 | 5.47 |
| | RTN | 55.63 | 51.80 | 34.54 | 30.01 | 48.70 | 27.40 | 26.11 | 25.00 | 3.98 | 33.69 | 1753.70 |
| | GPTQ | 54.95 | 54.13 | 35.88 | 31.60 | 49.88 | 32.66 | 25.26 | 25.60 | 4.08 | 34.89 | 9787.38 |
| **llama2-7b** | QuaRot | 74.13 | 77.09 | 43.50 | 73.27 | 66.85 | 70.24 | 43.52 | 39.60 | 70.10 | 62.03 | 6.12 |
| | Spinquant | 73.03 | 76.44 | 42.99 | 72.87 | 64.40 | 70.29 | 42.58 | 41.40 | 70.39 | 61.60 | 5.98 |
| | OSTQuant | 74.31 | 76.88 | 44.27 | 73.78 | 66.85 | 69.23 | 41.72 | 42.80 | 71.05 | 62.32 | 5.95 |
| | FlatQuant | 73.85 | 76.88 | 44.17 | 72.34 | 65.35 | 71.34 | 42.49 | 41.40 | 69.88 | 61.97 | 6.08 |
| | **ReSpinQuant** | **74.89** | **76.17** | **43.81** | **73.38** | **66.22** | **70.20** | **43.94** | **42.80** | **70.97** | **62.49** | **5.74** |
| | Full precision | 80.67 | 80.63 | 47.34 | 79.36 | 72.53 | 77.61 | 49.32 | 45.80 | 76.63 | 67.77 | 4.88 |
| | RTN | 39.54 | 51.90 | 35.01 | 26.91 | 47.75 | 29.92 | 23.98 | 25.80 | 0.31 | 31.24 | 3575.44 |
| | GPTQ | 38.47 | 52.45 | 34.80 | 27.51 | 50.51 | 31.27 | 23.89 | 24.20 | 0.04 | 31.46 | 2841.03 |
| **llama2-13b** | QuaRot | 79.27 | 78.18 | 44.98 | 76.19 | 69.53 | 75.46 | 45.82 | 44.00 | 73.80 | 65.25 | 5.37 |
| | Spinquant | 77.80 | 79.05 | 45.45 | 76.56 | 70.32 | 74.79 | 48.21 | 44.40 | 73.67 | 65.58 | 5.30 |
| | OSTQuant | 78.62 | 78.89 | 45.85 | 77.04 | 69.14 | 73.36 | 48.29 | 43.00 | 74.48 | 65.41 | 5.28 |
| | FlatQuant | 75.32 | 78.67 | 46.01 | 76.76 | 68.90 | 74.83 | 47.27 | 43.40 | 74.19 | 65.04 | 5.32 |
| | **ReSpinQuant** | **79.42** | **78.94** | **45.85** | **77.24** | **69.77** | **74.87** | **46.76** | **45.00** | **74.52** | **65.82** | **5.13** |
| | Full precision | 81.04 | 80.79 | 47.08 | 79.15 | 72.85 | 77.69 | 53.33 | 44.80 | 75.70 | 68.05 | 6.13 |
| | RTN | 45.05 | 57.83 | 36.28 | 38.38 | 51.46 | 36.20 | 24.66 | 30.20 | 10.62 | 36.74 | 219.82 |
| | GPTQ | 42.35 | 58.87 | 35.31 | 36.59 | 50.67 | 37.84 | 25.34 | 27.40 | 7.28 | 35.74 | 187.49 |
| **llama3-8b** | QuaRot | 76.61 | 77.58 | 44.93 | 74.83 | 67.48 | 67.59 | 44.80 | 43.00 | 69.26 | 62.90 | 7.82 |
| | Spinquant | 76.79 | 78.29 | 45.14 | 75.17 | 69.53 | 75.13 | 48.29 | 42.00 | 70.39 | 64.53 | 7.50 |
| | OSTQuant | 77.83 | 77.15 | 44.98 | 74.44 | 69.38 | 75.55 | 47.10 | 41.80 | 70.91 | 64.35 | 7.42 |
| | FlatQuant | 75.47 | 77.04 | 43.91 | 74.25 | 68.19 | 70.75 | 43.94 | 41.40 | 69.53 | 62.72 | 7.73 |
| | **ReSpinQuant** | **77.49** | **79.11** | **44.37** | **75.55** | **68.11** | **75.84** | **47.10** | **43.00** | **71.32** | **64.65** | **7.24** |
| | Full precision | 63.76 | 74.32 | 43.09 | 63.64 | 60.93 | 60.52 | 36.35 | 37.20 | 61.96 | 55.75 | 9.76 |
| | RTN | 51.01 | 55.55 | 33.98 | 31.09 | 49.88 | 30.09 | 23.63 | 24.80 | 1.65 | 33.52 | 330.79 |
| | GPTQ | 50.06 | 54.24 | 34.39 | 30.72 | 53.59 | 30.51 | 23.55 | 25.00 | 2.21 | 33.81 | 223.08 |
| **llama3.2-1b** | QuaRot | 50.83 | 67.41 | 39.25 | 54.23 | 52.96 | 50.25 | 30.55 | 31.20 | 44.61 | 46.81 | 14.59 |
| | Spinquant | 53.46 | 68.23 | 40.53 | 55.73 | 53.91 | 51.60 | 31.91 | 33.20 | 46.59 | 48.35 | 13.52 |
| | OSTQuant | 57.77 | 69.37 | 40.02 | 55.71 | 53.59 | 52.02 | 31.14 | 34.00 | 45.59 | 48.80 | 13.23 |
| | FlatQuant | 57.43 | 67.74 | 39.46 | 55.34 | 56.27 | 50.04 | 31.06 | 31.40 | 45.06 | 48.20 | 13.63 |
| | **ReSpinQuant** | **58.04** | **69.26** | **40.12** | **55.17** | **57.38** | **52.65** | **31.91** | **34.80** | **46.87** | **49.57** | **13.03** |
| | Full precision | 72.75 | 77.53 | 47.03 | 73.61 | 69.30 | 71.59 | 45.90 | 43.20 | 70.04 | 63.44 | 7.81 |
| | RTN | 47.34 | 56.31 | 35.21 | 36.36 | 49.72 | 34.34 | 25.85 | 28.00 | 3.18 | 35.15 | 266.80 |
| | GPTQ | 46.54 | 57.83 | 36.44 | 38.03 | 50.59 | 38.38 | 25.94 | 28.00 | 6.87 | 36.51 | 168.41 |
| **llama3.2-3b** | QuaRot | 66.42 | 73.12 | 44.47 | 67.67 | 63.06 | 63.64 | 38.99 | 38.80 | 58.78 | 57.22 | 9.99 |
| | Spinquant | 63.67 | 73.72 | 43.09 | 67.95 | 62.67 | 61.62 | 37.46 | 37.80 | 63.03 | 56.78 | 9.46 |
| | OSTQuant | 66.39 | 74.70 | 45.14 | 69.31 | 63.06 | 66.37 | 41.47 | 38.80 | 62.82 | 58.67 | 9.27 |
| | FlatQuant | 65.75 | 73.07 | 44.37 | 68.17 | 65.11 | 66.33 | 40.27 | 37.00 | 62.49 | 58.06 | 9.57 |
| | **ReSpinQuant** | **69.88** | **75.95** | **43.81** | **68.64** | **63.54** | **66.20** | **40.53** | **39.40** | **61.58** | **58.84** | **9.06** |

*Table 10.* Detailed W3A3 quantization results across benchmarks.

| Model | Method | BoolQ | PIQA | SIQA | HellaS. | WinoG. | ARC-e | ARC-c | OBQA | LAMB | Avg. (↑) | PPL (↓) |
|---|---|---|---|---|---|---|---|---|---|---|---|---|
| | Full precision | 77.77 | 79.05 | 45.96 | 76.00 | 68.51 | 74.54 | 46.16 | 44.00 | 73.70 | 65.08 | 5.47 |
| | RTN | 54.74 | 49.78 | 34.75 | 26.39 | 47.12 | 25.59 | 29.10 | 25.60 | 0.00 | 32.56 | 19519.63 |
| | GPTQ | 61.87 | 48.26 | 34.08 | 26.10 | 50.51 | 25.55 | 30.29 | 27.40 | 0.00 | 33.78 | 99691.84 |
| **llama2-7b** | QuaRot | 56.36 | 61.37 | 35.36 | 41.73 | 48.38 | 43.22 | 27.13 | 26.80 | 28.08 | 40.94 | 19.26 |
| | Spinquant | 62.60 | 67.79 | 38.43 | 56.99 | 54.78 | 50.84 | 32.51 | 32.00 | 46.42 | 49.15 | 9.69 |
| | OSTQuant | 60.12 | 68.88 | 38.54 | 56.14 | 55.56 | 52.23 | 29.95 | 30.80 | 43.78 | 48.44 | 9.53 |
| | FlatQuant | 58.07 | 63.38 | 38.08 | 49.80 | 55.72 | 49.62 | 28.24 | 29.40 | 36.60 | 45.43 | 13.32 |
| | **ReSpinQuant** | **65.84** | **67.95** | **39.41** | **59.52** | **55.72** | **49.41** | **29.78** | **31.20** | **52.47** | **50.14** | **8.57** |
| | Full precision | 80.67 | 80.63 | 47.34 | 79.36 | 72.53 | 77.61 | 49.32 | 45.80 | 76.63 | 67.77 | 4.88 |
| | RTN | 48.17 | 49.89 | 35.57 | 26.14 | 49.88 | 26.01 | 29.44 | 28.20 | 0.00 | 32.59 | 14960.35 |
| | GPTQ | 41.22 | 50.44 | 33.67 | 26.06 | 53.12 | 26.26 | 29.01 | 26.80 | 0.00 | 31.84 | 6837.00 |
| **llama2-13b** | QuaRot | 58.35 | 64.04 | 38.08 | 45.84 | 51.54 | 47.69 | 27.30 | 27.60 | 32.60 | 43.67 | 12.78 |
| | Spinquant | 68.20 | 69.97 | 41.71 | 65.38 | 60.14 | 59.34 | 36.86 | 35.20 | 59.17 | 55.10 | 7.36 |
| | OSTQuant | 69.27 | 72.31 | 41.86 | 66.48 | 61.09 | 59.22 | 34.90 | 37.00 | 65.11 | 56.36 | 7.10 |
| | FlatQuant | 63.24 | 69.91 | 39.15 | 57.38 | 57.46 | 55.05 | 32.85 | 32.20 | 51.31 | 50.95 | 9.27 |
| | **ReSpinQuant** | **70.46** | **72.42** | **42.02** | **67.20** | **59.59** | **60.23** | **36.77** | **37.40** | **63.13** | **56.58** | **6.91** |
| | Full precision | 81.04 | 80.79 | 47.08 | 79.15 | 72.85 | 77.69 | 53.33 | 44.80 | 75.70 | 68.05 | 6.13 |
| | RTN | 47.95 | 52.23 | 33.06 | 26.04 | 52.41 | 25.59 | 26.71 | 30.40 | 0.02 | 32.71 | 77055.20 |
| | GPTQ | 47.49 | 50.54 | 32.45 | 26.93 | 49.88 | 26.60 | 26.11 | 30.80 | 0.00 | 32.31 | 44824.84 |
| **llama3-8b** | QuaRot | 50.34 | 52.94 | 33.78 | 31.68 | 51.30 | 31.23 | 22.53 | 24.80 | 6.99 | 33.95 | 98.04 |
| | Spinquant | 61.22 | 66.21 | 37.62 | 56.48 | 55.72 | 52.95 | 30.97 | 33.00 | 44.75 | 48.77 | 15.07 |
| | OSTQuant | 60.15 | 65.13 | 38.02 | 56.04 | 55.01 | 47.31 | 29.78 | 32.20 | 42.46 | 47.34 | 14.29 |
| | FlatQuant | 47.22 | 53.81 | 34.85 | 32.23 | 51.38 | 31.19 | 24.15 | 24.20 | 5.26 | 33.81 | 133.52 |
| | **ReSpinQuant** | **65.75** | **67.41** | **40.07** | **59.86** | **55.88** | **54.63** | **31.91** | **35.60** | **45.59** | **50.74** | **13.09** |
| | Full precision | 63.76 | 74.32 | 43.09 | 63.64 | 60.93 | 60.52 | 36.35 | 37.20 | 61.96 | 55.75 | 9.76 |
| | RTN | 42.35 | 50.44 | 33.57 | 26.48 | 51.85 | 24.83 | 27.47 | 27.60 | 0.00 | 31.62 | 115358.15 |
| | GPTQ | 42.02 | 50.22 | 32.91 | 26.02 | 49.17 | 25.88 | 26.19 | 27.80 | 0.00 | 31.13 | 56762.62 |
| **llama3.2-1b** | QuaRot | 49.94 | 50.65 | 33.52 | 27.84 | 49.57 | 27.99 | 23.29 | 25.00 | 0.87 | 32.07 | 812.46 |
| | Spinquant | 51.74 | 54.57 | 34.19 | 32.84 | 50.12 | 32.87 | 23.12 | 29.00 | 6.75 | 35.02 | 69.70 |
| | OSTQuant | 53.52 | 55.28 | 34.70 | 32.46 | 50.04 | 36.20 | 24.15 | 26.40 | 8.64 | 35.71 | 59.71 |
| | FlatQuant | 49.20 | 52.61 | 32.86 | 27.63 | 50.28 | 29.42 | 23.89 | 25.20 | 0.62 | 32.41 | 543.66 |
| | **ReSpinQuant** | **56.48** | **58.00** | **34.65** | **35.04** | **50.67** | **36.74** | **21.76** | **24.60** | **11.95** | **36.65** | **49.90** |
| | Full precision | 72.75 | 77.53 | 47.03 | 73.61 | 69.30 | 71.59 | 45.90 | 43.20 | 70.04 | 63.44 | 7.81 |
| | RTN | 41.25 | 51.96 | 34.49 | 25.78 | 51.30 | 24.75 | 24.83 | 27.20 | 0.00 | 31.28 | 22132.04 |
| | GPTQ | 40.55 | 51.69 | 33.32 | 25.95 | 51.30 | 26.43 | 24.83 | 29.20 | 0.02 | 31.48 | 10753.82 |
| **llama3.2-3b** | QuaRot | 44.43 | 55.11 | 33.32 | 30.67 | 51.38 | 29.46 | 22.10 | 25.80 | 4.17 | 32.94 | 172.33 |
| | Spinquant | 52.29 | 60.50 | 37.51 | 47.37 | 52.09 | 43.06 | 27.47 | 29.40 | 29.71 | 42.16 | 21.05 |
| | OSTQuant | 53.43 | 61.21 | 38.23 | 42.80 | 52.64 | 42.00 | 26.11 | 29.20 | 24.82 | 41.16 | 21.27 |
| | FlatQuant | 50.40 | 54.24 | 34.34 | 32.06 | 51.54 | 30.98 | 23.63 | 24.80 | 5.14 | 34.13 | 84.47 |
| | **ReSpinQuant** | **61.47** | **60.88** | **36.49** | **48.31** | **52.09** | **45.12** | **27.99** | **28.60** | **31.71** | **43.63** | **18.61** |

