# OpenReview forum: "ReSpinQuant: Efficient Layer-Wise LLM Quantization via Subspace Residual Rotation Approximation"
_ICML.cc/2026/Conference — ICML 2026 regular_

### Official Review · Reviewer_tX74 · 2026-03-08

**Soundness:** 2
**Presentation:** 3
**Significance:** 3
**Originality:** 2
**Overall Recommendation:** 4
**Confidence:** 3

**Summary:**

This paper focuses on rotation-based PTQ methods and introduces ReSpinQuant, which reduces the significant computational overhead by using layer-wise orthogonal matrices that are mostly fused into the weights offline. ReSpinQuant is evaluated on LLaMA models, showing improvements in W4A4 and W3A3 settings.

**Compliance With Llm Reviewing Policy:**

Affirmed.

**Final Justification:**

This paper introduces ReSpinQuant, a rotation-based PTQ method that reduces computational overhead by fusing layer-wise orthogonal matrices offline, showing empirical improvements in W4A4 and W3A3 settings.

I thank the authors for their rebuttal, which successfully addressed my initial concerns regarding memory bottlenecks and learning rate stability.

Overall, I have a positive view of this paper.

**Key Questions For Authors:**

- Can the authors acknowledge the memory footprint and bandwidth overhead associated with storing and loading layer-wise projection matrices during inference.
- Can the authors provide experimental results under different learning rates and compare them with SpinQuant?
- Can the authors discuss the potential confounding effect of using 1091M parameters during calibration, as this may limit the ability to definitively attribute the performance gains solely to the architectural design?
- The key approximation is based on empirical observations, can the authors analyze the theoretical bounds of this approximation?

**Limitations:**

yes

**Strengths And Weaknesses:**

Strengths:
- Reduces online residual alignment overhead via subspace approximation.
- Fuses layer-wise rotations into weight matrices offline for zero-overhead inference.
- Achieves robust W4A4 and W3A3 PTQ performance on LLaMA-2 and LLaMA-3 models.

Weakness:
- Layer-wise projections create memory bottlenecks that likely negate the hardware latency gains.
- The 63× increase in calibration parameters requires matched-budget ablation studies for a fair comparison.
- The proposed framework presents an incremental combination of existing engineering techniques.
- The core empirical approximation lacks mathematical justification.

---

> ### Author Rebuttal · Authors · 2026-03-31
>
> Thank you for the careful review. We appreciate that you recognized the strong empirical performance of the paper and the importance of the efficiency–expressivity trade-off in rotation-based PTQ. We address your concerns below.
>
> > **Weakness 1: Layer-wise projections create memory bottlenecks that likely negate the hardware latency gains.**
>
> > **Q1: Can the authors acknowledge the memory footprint and bandwidth overhead associated with storing and loading layer-wise projection matrices during inference.**
>
> We agree that online residual correction introduces additional memory/bandwidth overhead. However, we respectfully disagree that this overhead “likely negates” the hardware latency gains.
>
> The key point is that the low-rank residual correction accounts for only a small fraction of the total memory footprint. In our additional measurement on LLaMA-3 8B W4A4KV4, the total memory per layer is **105.0 MB**, of which only **1.0 MB** comes from the low-rank residual correction, while **104.0 MB** comes from the model weights. This is consistent with the end-to-end latency results already reported in Table 3 of the paper: on H100, the TTIT at batch size 16 increases only from **160.95 ms** to **163.81 ms**, i.e., about **1.7%**. Thus, the extra bandwidth cost does exist, but it does not create a memory bottleneck.
>
> > **Q2: Can the authors provide experimental results under different learning rates and compare them with SpinQuant?**
>
> Thank you for this suggestion. We ran an additional learning-rate sweep for ReSpinQuant, using the same overall calibration protocol and varying only the learning rate:
>
> |lr|Llama2-7B(PPL)|Llama2-7B(zeroshot)|Llama3-8B(PPL)|Llama3-8B(zeroshot)|
> |:---:|:---:|:---:|:---:|:---:|
> |1.5|5.98|61.79|7.37|64.37|
> |5|5.82|61.81|7.24|64.93|
> |10|5.76|61.46|7.22|64.36|
> |15|5.98|61.60|7.24|64.65|
> |30|5.83|61.72|7.52|64.24|
> |45|6.07|61.30|7.98|62.42|
> |60|6.37|60.26|9.50|59.30|
>
> These results show that ReSpinQuant is stable across a broad range of learning rates (1.5–30) and degrades clearly only at very large values such as 45 and 60. In the main paper, we used lr=15 as a single default across models rather than tuning per model.
>
> > **Weakness 2: The 63× increase in calibration parameters requires matched-budget ablation studies for a fair comparison.**
>
> > **Q3: Can the authors discuss the potential confounding effect of using 1091M parameters during calibration, as this may limit the ability to definitively attribute the performance gains solely to the architectural design?**
>
> We agree that ReSpinQuant uses a larger calibration-time parameterization than SpinQuant (shown in Table 4 of the main paper). Our claim is not about better offline efficiency. Instead, ReSpinQuant is deliberately designed around a 'train-large, infer-small' paradigm. We spend more offline capacity during calibration, but most of it is fused away before deployment, leaving only a small online overhead. This design choice is highly practical, as calibration is a one-time cost, whereas inference is executed continuously. Fundamentally, enabling the use of more calibration parameters with only small online overhead is the core of our architectural design.
>
> > **Weakness 4 and Q4: The core empirical approximation lacks mathematical justification.**
>
> > **Q4: The key approximation is based on empirical observations, can the authors analyze the theoretical bounds of this approximation?**
>
> We agree that the original manuscript emphasized empirical evidence more than formal analysis. Our claim is not that the low-rank correction is exact, but that the residual transition T is a small deviation around identity and is highly compressible. To quantify this, we additionally measured the cumulative energy of (T-I) (averaged across layers) on LLaMA-2 7B W3A3:
>
> |Rank|0|8|16|32|64|128|4096|
> |:---|:---:|:---:|:---:|:---:|:---:|:---:|:---:|
> |**Cumulative Energy**|0%|51.07%|53.45%|**55.77%**|58.55%|62.33%|100%|
> |**$\Vert\hat{T}_r-I\Vert_F/\Vert T-I\Vert_F$**|0%|70.25%|72.09%|**73.80%**|75.76%|78.34%|100%|
>
>
> With **rank 32**, we use only **32/4096 = 0.78%** of the full dimension, yet still capture **55.77%** of the cumulative energy of (T-I) and retain **73.8%** of the Frobenius norm of the residual correction. This suggests that the basis mismatch is concentrated in a low-dimensional subspace. Consistent with this, T remains close to the identity, and the residual term T-I is well captured by its top-r singular subspace.
>
> We hope these additional analyses and experiments clarify the reviewer’s concerns. We would be happy to further discuss these points and incorporate the suggested clarifications in the revision.

---

> > ### Author Rebuttal · Reviewer_tX74 · 2026-04-03
> >
> > Thank the authors for the detailed rebuttal. My concerns regarding the memory bottleneck and learning rate stability have been addressed. However, I still have concerns about the fairness of the comparison using 63× calibration parameters, as well as the lack of theoretical justification.
> >
> > Can the authors include a comparison using the same learning rate settings for SpinQuant to enable a direct comparison? Currently, only the learning rate sensitivity of ReSpinQuant is provided, and it remains unclear whether the underperformance of SpinQuant stems from the use of a smaller learning rate.
> >
> > Can the authors provide detailed matched-budget ablation studies for a fair comparison? This would help isolate the contribution of each component in ReSpinQuant and identify the primary source of performance gains.
> >
> > Finally, the core approximation still lacks rigorous theoretical bound analysis.
> >
> > I am willing to increase my score if these concerns are adequately addressed.

---

> > > ### Author Response · Authors · 2026-04-07
> > >
> > > We appreciate your confirmation that the concerns regarding the memory bottleneck and learning rate stability have been resolved. Below are our responses to the remaining unaddressed concerns.
> > >
> > > > **Q1: Learning Rate Comparison to SpinQuant**
> > >
> > > We provide a direct comparison between SpinQuant and ReSpinQuant across different learning rates.
> > >
> > > |Method(lr)|Llama2-7B (PPL)|Llama2-7B (zeroshot)|Llama2-13B (PPL)|Llama2-13B (zeroshot)|Llama3-8B (PPL)|Llama3-8B (zeroshot)|Llama3.2-1B (PPL)|Llama3.2-1B (zeroshot)|Llama3.2-3B (PPL)|Llama3.2-3B (zeroshot)|**average ppl**|**average zeroshot**|
> > > |:---|:---:|:---:|:---:|:---:|:---:|:---:|:---:|:---:|:---:|:---:|:---:|:---:|
> > > |**Spinquant(1.5)**|5.98|61.60|5.30|65.58|7.50|64.53|13.52|48.35|9.46|56.78|**8.35**|**59.37**|
> > > |**Spinquant(5)**|5.95|61.71|5.26|65.26|7.40|63.69|13.36|49.01|9.20|57.70|**8.23**|**59.47**|
> > > |**Spinquant(10)**|5.94|61.16|5.26|65.22|7.40|64.48|13.71|49.19|9.21|57.48|**8.30**|**59.51**|
> > > |**Spinquant(15)**|5.92|61.20|5.25|65.67|7.47|64.08|13.31|48.40|9.31|57.69|**8.25**|**59.41**|
> > > ||||||||||||||
> > > |**ReSpinQuant(1.5)**|5.98|61.79|5.27|65.30|7.37|64.37|13.08|48.98|9.17|57.77|**8.17**|**59.64**|
> > > |**ReSpinQuant(5)**|5.83|61.81|5.14|65.90|7.25|64.93|12.91|48.70|8.98|57.90|**8.02**|**59.85**|
> > > |**ReSpinQuant(10)**|5.77|61.64|5.11|65.28|7.23|64.36|13.34|48.07|8.94|58.51|**8.08**|**59.57**|
> > > |**ReSpinQuant(15)**|5.74|62.49|5.12|65.78|7.24|64.65|13.03|49.57|9.06|58.84|**8.04**|**60.27**|
> > >
> > > As the results indicate, ReSpinQuant consistently outperforms at any given learning rate, while simultaneously exhibiting a more significant performance improvement at higher learning rates. We attribute this to ReSpinQuant having a larger number of learnable parameters during the calibration phase, making it more capable of absorbing updates and thus more suited for higher learning rates.
> > >
> > > > **Q2: Matched-Budget Ablation**
> > >
> > > Regarding the concern that our method utilizes 63x more calibration parameters due to layer-wise rotation matrices compared to SpinQuant's global rotation matrix, we would like to respectfully clarify that this is a core strength of our approach, not a weakness.
> > >
> > > Structurally, SpinQuant cannot scale its parameter count without directly imposing a severe inference overhead. In contrast, ReSpinQuant intentionally leverages a larger layer-wise parameter space during the calibration phase to achieve superior performance under extreme quantization settings. Crucially, we maintain low overhead during inference by approximating these matrices with a low-rank structure. Because decoupling the high-capacity calibration space from the low-overhead inference space is the fundamental design principle of ReSpinQuant, a direct matched-budget ablation is structurally infeasible and contradicts the premise of our method.
> > >
> > > > **Q3: Theoretical Bound Analysis of Approximation**
> > >
> > > We can formally prove that our low-rank approximated rotation matrix is the mathematically optimal low-rank rotation approximation, and that its error is strictly bounded by the tail singular values.
> > >
> > > The approximated transition matrix $\hat{\mathbf{T}}$ is formulated as:
> > > $$\hat{\mathbf{T}} = \mathbf{I} + \mathbf{Q}(\hat{\mathbf{R}}_{\text{sub}} - \mathbf{I}_r)\mathbf{Q}^T$$
> > > $$\hat{\mathbf{T}} - \mathbf{I} = \mathbf{Q}(\mathbf{Q}^T \mathbf{T} \mathbf{Q} - \mathbf{Q}^T \mathbf{I} \mathbf{Q})\mathbf{Q}^T$$
> > > $$\hat{\mathbf{T}} - \mathbf{I} = \mathbf{Q}(\mathbf{Q}^T \Delta \mathbf{T} \mathbf{Q})\mathbf{Q}^T$$
> > >
> > > By definition of SVD, $\mathbf{Q}(\mathbf{Q}^T \Delta \mathbf{T} \mathbf{Q})\mathbf{Q}^T$ is the optimal rank-$r$ approximation of $\Delta \mathbf{T}$.
> > > $$\therefore \hat{\mathbf{T}} - \mathbf{I} = \Delta \mathbf{T}_r$$
> > > $$\|\mathbf{T} - \hat{\mathbf{T}}\|_F = \|(\mathbf{I} + \Delta \mathbf{T}) - (\mathbf{I} + \Delta \mathbf{T}_r)\|_F = \|\Delta \mathbf{T} - \Delta \mathbf{T}_r\|_F$$
> > >
> > > By the Eckart-Young-Mirsky Theorem, for a matrix $\Delta \mathbf{T}$ with singular values $\sigma_1 \ge \sigma_2 \ge \dots \ge \sigma_D$:
> > > $$\|\Delta \mathbf{T} - \Delta \mathbf{T}\_r\|\_F = \sqrt{\sum\_{i=r+1}^D \sigma\_i^2}$$
> > > $$\therefore \|\mathbf{T} - \hat{\mathbf{T}}\|\_F = \sqrt{\sum\_{i=r+1}^D \sigma\_i^2} \le \|\mathbf{T} - \mathbf{R}\|\_F$$
> > > for any $\mathbf{R} \in SO(D)$ such that $\text{rank}(\mathbf{R} - \mathbf{I}) \le r$
> > >
> > > Given the empirical observation that high singular values strongly dominate the transition matrix, this proof formally guarantees why our low-rank approximation is highly effective with a negligible performance drop.
> > >
> > > We hope these explanations and theoretical proofs successfully resolve your remaining concerns.

---

### Official Review · Reviewer_f6A8 · 2026-03-12

**Soundness:** 3
**Presentation:** 3
**Significance:** 3
**Originality:** 3
**Overall Recommendation:** 4
**Confidence:** 4

**Summary:**

This paper studies post-training quantization for large language models and targets the well-known tradeoff between expressive layer-wise rotations and efficient global rotations. The proposed method, ReSpinQuant, learns layer-specific rotations during calibration, fuses the main transformations into the weights offline, and then approximates the remaining residual basis mismatch with a low-rank subspace correction. The goal is to retain most of the accuracy benefits of layer-wise adaptation while keeping inference overhead close to efficient global-rotation methods. The paper reports strong results on low-bit settings such as W4A4 and W3A3, together with latency measurements suggesting only modest runtime overhead.

**Compliance With Llm Reviewing Policy:**

Affirmed.

**Key Questions For Authors:**

1. The paper already compares against OSTQuant[4], but can you provide either direct experiments or a careful analytical comparison against additional accepted recent PTQ methods such as DuQuant[5] and KurTail[6]? A stronger answer here would materially increase my confidence in the paper's significance.
2. Please clarify the end-to-end calibration and inference pipeline. How exactly are the low-rank subspace basis and residual rotation parameters obtained, and what computations remain online after offline weight fusion?
3. How robust is the core low-rank residual assumption across all layers, different calibration sets, and larger models? It would help to show layer-wise statistics, sensitivity to calibration data, and rank-vs-performance behavior beyond the current limited ablation.
4. The method appears especially strong in very low-bit settings such as W3A3. Can you provide more intuition and analysis for why the layer-wise residual correction matters more there, and when you would expect the gains over simpler rotation baselines to diminish?

References
1. SpinQuant: LLM Quantization with Learned Rotations. Zechun Liu, Changsheng Zhao, Igor Fedorov, Bilge Soran, Dhruv Choudhary, Raghuraman Krishnamoorthi, Vikas Chandra, Yuandong Tian, Tijmen Blankevoort. ICLR 2025. https://openreview.net/forum?id=ogO6DGE6FZ
2. QuaRot: Outlier-Free 4-Bit Inference in Rotated LLMs. Saleh Ashkboos, Amirkeivan Mohtashami, Maximilian L. Croci, Bo Li, Martin Jaggi, Dan Alistarh, Torsten Hoefler, James Hensman. NeurIPS 2024. https://arxiv.org/abs/2404.00456
3. FlatQuant: Flatness Matters for LLM Quantization. Yuxuan Sun, Chenhao Lu, Jianli Jiao, Yanjing Li, et al. ICLR 2025. https://proceedings.mlr.press/v267/sun25l.html
4. OSTQuant: Refining Large Language Model Quantization with Orthogonal and Scaling Transformations for Better Distribution Fitting. Zhongyu Xie, Xintao Wang, Hongliang Zhong, Chao Dong, Ying Shan. ICLR 2025. https://openreview.net/forum?id=rAcgDBdKnP
5. DuQuant: Distributing Outliers via Dual Transformation Makes Stronger Quantized LLMs. Haokun Lin, Haobo Xu, et al. ICLR 2025. https://openreview.net/forum?id=mp8u2Pcmqz
6. KurTail: Kurtosis-based LLM Quantization. Yongqi An, et al. EMNLP 2025 Findings. https://aclanthology.org/2025.findings-emnlp.943/

**Limitations:**

yes

**Strengths And Weaknesses:**

Strengths: The central idea is well motivated: prior rotation-based PTQ methods such as SpinQuant[1] and QuaRot[2] either share one transformation across layers and lose expressivity, while stronger layer-wise alternatives such as FlatQuant[3] and OSTQuant[4] pay substantially more online cost or structural restrictions. ReSpinQuant offers a plausible middle ground through a low-rank residual parameterization. The paper also already includes direct empirical comparisons against SpinQuant[1], QuaRot[2], OSTQuant[4], and FlatQuant[3], which strengthens the experimental case. The empirical results are strong, especially in aggressive low-bit settings, and the efficiency story is one of the paper's main strengths. In particular, the paper appears to show that most of the expensive transformation can be absorbed offline, leaving only a lightweight online correction.

Weaknesses: The paper's explanation of the basis-mismatch problem, the calibration pipeline, and the exact inference computation flow is still harder to follow than it should be. It remains somewhat unclear how the low-rank subspace objects are learned in practice, how sensitive the method is to calibration data, and how robust the near-identity / low-rank residual assumption is across layers, model scales, and deployment distributions. The evaluation is promising, but I would like more complete efficiency reporting beyond a narrow latency setup, including memory footprint and throughput across multiple sequence lengths or decoding regimes. Scalability evidence beyond the currently tested scale also remains limited.

Concerns: The paper does not yet position itself strongly enough against the broader recent SOTA landscape. The paper already compares against several strong baselines, including SpinQuant[1], QuaRot[2], OSTQuant[4], and FlatQuant[3], which is a meaningful strength. However, I would still like a more careful comparison or analytical discussion relative to additional accepted rotation- or transformation-based PTQ methods discussed in the related-work survey, especially DuQuant[5] and KurTail[6]. If these are not directly comparable because of different settings, the paper should explain the tradeoffs clearly. A stronger discussion of how ReSpinQuant differs from, or improves on, these recent approaches would materially strengthen the significance and originality claims. As written, the paper looks technically solid and useful, but the comparative positioning can still be strengthened.

References
1. SpinQuant: LLM Quantization with Learned Rotations. Zechun Liu, Changsheng Zhao, Igor Fedorov, Bilge Soran, Dhruv Choudhary, Raghuraman Krishnamoorthi, Vikas Chandra, Yuandong Tian, Tijmen Blankevoort. ICLR 2025. https://openreview.net/forum?id=ogO6DGE6FZ
2. QuaRot: Outlier-Free 4-Bit Inference in Rotated LLMs. Saleh Ashkboos, Amirkeivan Mohtashami, Maximilian L. Croci, Bo Li, Martin Jaggi, Dan Alistarh, Torsten Hoefler, James Hensman. NeurIPS 2024. https://arxiv.org/abs/2404.00456
3. FlatQuant: Flatness Matters for LLM Quantization. Yuxuan Sun, Chenhao Lu, Jianli Jiao, Yanjing Li, et al. ICLR 2025. https://proceedings.mlr.press/v267/sun25l.html
4. OSTQuant: Refining Large Language Model Quantization with Orthogonal and Scaling Transformations for Better Distribution Fitting. Zhongyu Xie, Xintao Wang, Hongliang Zhong, Chao Dong, Ying Shan. ICLR 2025. https://openreview.net/forum?id=rAcgDBdKnP
5. DuQuant: Distributing Outliers via Dual Transformation Makes Stronger Quantized LLMs. Haokun Lin, Haobo Xu, et al. ICLR 2025. https://openreview.net/forum?id=mp8u2Pcmqz
6. KurTail: Kurtosis-based LLM Quantization. Yongqi An, et al. EMNLP 2025 Findings. https://aclanthology.org/2025.findings-emnlp.943/

---

> ### Author Rebuttal · Authors · 2026-03-31
>
> Thank you for the detailed review and the positive assessment. We appreciate that you found the idea well-motivated and the empirical results strong. Below we address your concerns.
>
> > **Q1: Can you compare against DuQuant/KurTail?**
>
> We already compare against SpinQuant, QuaRot, OSTQuant, and FlatQuant. To broaden the comparison, we additionally ran DuQuant under the same W4A4KV4 setting. ReSpinQuant outperforms DuQuant on all 10 reported metrics:
>
> |**W4A4KV4**|Llama2-7B (PPL)|Llama2-7B (zeroshot)|Llama2-13B (PPL)|Llama2-13B (zeroshot)|Llama3-8B (PPL)|Llama3-8B (zeroshot)|Llama3.2-1B (PPL)|Llama3.2-1B (zeroshot)|Llama3.2-3B (PPL)|Llama3.2-3B (zeroshot)|
> |:---|:---:|:---:|:---:|:---:|:---:|:---:|:---:|:---:|:---:|:---:|
> |**DuQuant**|6.09 / 6.08*|61.78|5.33 / 5.33*|65.06|8.63 / 8.06*|59.99|22.24|42.17|10.35|55.42|
> |**ReSpinQuant (ours)**|**5.74**|**62.49**|**5.12**|**65.78**|**7.24**|**64.65**|**13.03**|**49.57**|**9.06**|**58.84**|
>
> We reproduced the DuQuant baseline using their official GitHub repository. The numbers reported in their original paper are marked with an asterisk (*). While there is a discrepancy between our reproduced results and their reported numbers for Llama3-8B, ReSpinQuant consistently outperforms DuQuant in both cases.
>
> For KurTail, we were unable to run a direct comparison because the code is not publicly available. More broadly, the two methods take fundamentally different approaches: KurTail mainly focuses on modifying the optimization and loss, whereas ReSpinQuant contributes an inference-efficient architecture for deploying dense layer-wise rotations.
>
> > **Q2: Please clarify the end-to-end calibration and inference pipeline.**
>
> A clearer separation between calibration and inference may help. During calibration, we learn full-rank layer-wise orthogonal rotations, and no low-rank approximation is used at this stage. After calibration, for each residual transition ($T=R_{\text{out}}R_{\text{in}}^\top$), we compute ($\Delta T=T-I$), extract the top-r subspace by SVD, project (T) into that subspace, and obtain the closest orthogonal ($r\times r$) matrix by polar decomposition. The dense full-rank rotations in MHSA/FFN are then fused offline into the weight matrices, while only the resulting low-rank residual correction remains online. This process is currently described around line 277 (right column) of the submission, but we will revise the text to make this end-to-end pipeline much clearer.
>
> > **Q3: How robust is the low-rank residual assumption across layers, calibration sets, and larger models?**
>
> To better quantify the low-rank assumption, we additionally measured the average spectral statistics of (T-I) on LLaMA-2 7B W3A3:
>
> |Rank|0|8|16|32|64|128|4096|
> |:---|:---:|:---:|:---:|:---:|:---:|:---:|:---:|
> |**Cumulative Energy**|0%|51.07%|53.45%|**55.77%**|58.55%|62.33%|100%|
> |**$\Vert\hat{T}_r-I\Vert_F/\Vert T-I\Vert_F$**|0%|70.25%|72.09%|**73.80%**|75.76%|78.34%|100%|
>
> At rank 32, we use only **0.78%** of the 4096-dim feature space, yet capture **55.77%** of the cumulative energy of (T-I) and retain **73.8%** of the Frobenius magnitude of the residual correction. This supports that the basis mismatch is highly compressible. We also observed layer-wise variation (std = 16.78% at r=32), suggesting adaptive rank selection as an interesting future direction.
>
> We also tested calibration sensitivity by replacing WikiText-2 with the C4 dataset. Zero-shot performance changed only marginally:
>
> |W4A4KV4 (C4)|Llama2-7B (PPL)|Llama2-7B (zeroshot)|Llama2-13B (PPL)|Llama2-13B (zeroshot)|Llama3-8B (PPL)|Llama3-8B (zeroshot)|Llama3.2-1B (PPL)|Llama3.2-1B (zeroshot)|Llama3.2-3B (PPL)|Llama3.2-3B (zeroshot)|
> |:---|:---:|:---:|:---:|:---:|:---:|:---:|:---:|:---:|:---:|:---:|
> |**SpinQuant**|8.07|61.67|7.39|65.46|11.88|63.79|20.33|49.11|14.39|58.01|
> |**ReSpinQuant**|**7.70**|**62.40**|**7.29**|**66.06**|**11.50**|**64.67**|**19.82**|**49.48**|**13.99**|**58.64**|
>
> This suggests that the method is not highly sensitive to the calibration corpus, demonstrating robust and consistent performance across different calibration datasets. We agree that experiments beyond the 13B scale would be valuable. However, we were unable to evaluate larger models due to GPU and time constraints.
>
> > **Q4: Why is the method especially strong in W3A3, and when do gains diminish?**
>
> In higher-bit settings, a single global rotation already keeps quantization noise small, so the extra expressivity of layer-wise adaptation brings only modest gains. In very low-bit regimes such as W4A4/W3A3, quantization becomes much more sensitive to outliers, so correcting layer-wise rotation matrices matters more. Accordingly, we observed the gain over simpler global-rotation baselines to diminish as the quantization setting becomes milder (e.g., W4A8KV8).
>
> Thank you again for the constructive feedback. We hope this additional information successfully resolves your concerns, and we would be happy to engage in further discussion.

---

### Official Review · Reviewer_SqF3 · 2026-03-13

**Soundness:** 3
**Presentation:** 3
**Significance:** 2
**Originality:** 2
**Overall Recommendation:** 4
**Confidence:** 3

**Summary:**

This paper studies rotation-based post-training quantization for LLMs. Global rotation methods are efficient because the rotations can be fused offline, but they are limited by using a single basis across layers. Layer-wise methods are more expressive, but they introduce substantial online overhead. The paper proposes `ReSpinQuant`, which uses layer-wise rotations while absorbing them into the weights offline, and handles the resulting basis mismatch in the residual path using a low-rank subspace residual rotation approximation. The method is evaluated on `LLaMA-2 / 3 / 3.2` models in both `W4A4` and `W3A3` settings, with results on perplexity, zero-shot accuracy, MACs, latency, and calibration cost.

**Compliance With Llm Reviewing Policy:**

Affirmed.

**Final Justification:**

According to the experimental results in the rebuttal, the improvement is mostly marginal in certain settings, which were not fully covered in the original submission. I am keeping the overall score.

**Key Questions For Authors:**

1. Can the authors better justify the low-rank residual approximation quantitatively, for example by reporting singular value decay of `T - I`, cumulative energy captured by the top-`r` components, or approximation errors for larger `r` / full-rank references?
2. How robust is the method on architectures beyond the `LLaMA` family and on other bit-width setups?

**Limitations:**

Yes.

**Strengths And Weaknesses:**

Strengths:

- The paper studies a clear and practically important question in rotation-based LLM quantization.
- The proposed design is easy to understand and the presentation is clear.
- The ablation on the approximation rank is useful and supports the choice of a low-rank residual correction.

Weaknesses:

- The key justification for the low-rank residual approximation is mainly empirical. The paper shows that the learned rotations remain close to the initial Hadamard structure, but the scope of this phenomenon is not yet well characterized. More specifically, the residual correction is an approximation rather than an equivalent reformulation, and the chosen rank is very small relative to the hidden dimension (e.g., `r=32` for features with several thousand dimensions). While the ablation suggests that this works empirically, the paper does not yet provide enough evidence that the residual mismatch is consistently concentrated in such a low-dimensional subspace, for example via singular value decay or approximation error statistics (instead of only reporting the PPL and Zero-shot Avg. and treat the error as a black box).
- The evaluation is limited. Prior studies like `SpinQuant` report the results of W-A-KV = 4-8-16, 4-8-8, 4-4-16, 4-4-4. However, this paper only report W-A = 4-4, 3-3. Could the authors justify their experiemntal setup or cover a broader range of bit-width? Notice that the scores reported are significantly worse than the original 16-16 models. Following the first bullet, the aggressive low-rank approximation does not seem to be generically applicable in theory. It should be supported by more experimental results.

---

> ### Author Rebuttal · Authors · 2026-03-31
>
> Thank you for the careful and detailed review. We appreciate your positive assessment and your highlighting of the practical importance of the problem. We think that the trade-off between fusible global rotations and expressive layer-wise transformations is one of the central challenges in rotation-based LLM PTQ. ReSpinQuant is designed precisely to bridge this gap: it preserves the representation power of layer-wise rotations while restricting the online overhead to a lightweight low-rank residual correction.
>
> Below we address your questions.
>
> > **Weakness 1 and Q1: Can the authors better justify the low-rank residual approximation quantitatively, for example by reporting singular value decay of (T-I), cumulative energy captured by the top-r components, or approximation errors for larger r / full-rank references?**
>
> Thank you for this suggestion. We agree that the original submission would benefit from a more quantitative low-rank justification. To directly address this point, we additionally measured the average cumulative energy of (T-I) across all layers on LLaMA-2 7B, W3A3, as well as the fraction of the Frobenius norm retained by the approximate residual correction:
>
> |Rank|0|8|16|32|64|128|4096|
> |:---|:---:|:---:|:---:|:---:|:---:|:---:|:---:|
> |**Cumulative Energy**|0%|51.07%|53.45%|**55.77%**|58.55%|62.33%|100%|
> |**$\Vert\hat{T}_r-I\Vert_F/\Vert T-I\Vert_F$**|0%|70.25%|72.09%|**73.80%**|75.76%|78.34%|100%|
>
> These results show that the residual mismatch is indeed concentrated in a much smaller subspace. In particular, with rank 32, we use only **32/4096 = 0.78%** of the full feature dimension, yet still capture **55.77%** of the cumulative energy of (T-I), and retain **73.80%** of the Frobenius magnitude of the full residual correction. This supports our view that preserving the dominant part of this residual is enough to recover most of the practical accuracy gain. This is also consistent with the rank ablation in Table 5 of the paper: higher ranks continue to improve the approximation, but rank 32 gives a strong accuracy-efficiency trade-off.
>
> > **Weakness 2 and Q2: The evaluation is limited. Prior studies like SpinQuant report the results of W-A-KV = 4-8-16, 4-8-8, 4-4-16, 4-4-4. However, this paper only reports W-A = 4-4, 3-3. Could the authors justify their experimental setup or cover a broader range of bit-width?**
>
> We would first like to clarify the notation. For brevity, we write W4A4 and W3A3, but in our experiments the KV cache is quantized to the same precision as the activations. Thus, these settings correspond to W4A4KV4 and W3A3KV3.
>
> We focused on W4A4KV4 and W3A3KV3 because these are the most challenging and practically informative regimes for our method. In higher-bit settings, strong baselines such as SpinQuant are already much closer to full precision, so the room for additional gains naturally becomes smaller. Since ReSpinQuant addresses the representational limitation of a single global basis, its benefit is expected to be most visible when the quantization budget is aggressive.
>
> To further address your concern, we additionally evaluated W4A8KV8 and W4A4KV16 on five LLaMA-family models:
>
> |Method|Llama2-7B (PPL)|Llama2-7B (zeroshot)|Llama2-13B (PPL)|Llama2-13B (zeroshot)|Llama3-8B (PPL)|Llama3-8B (zeroshot)|Llama3.2-1B (PPL)|Llama3.2-1B (zeroshot)|Llama3.2-3B (PPL)|Llama3.2-3B (zeroshot)|
> |:---|:---:|:---:|:---:|:---:|:---:|:---:|:---:|:---:|:---:|:---:|
> |**FP**|5.47|65.08|4.88|67.77|6.13|68.05|9.76|55.75|7.81|63.44|
> |**W4A8KV8 SpinQuant**|5.62|64.07|5.00|66.85|**6.53**|67.28|**10.5**|**53.39**|8.24|62.24|
> |**W4A8KV8 ReSpinQuant**|**5.60**|**64.24**|**4.98**|**67.10**|**6.53**|**67.32**|10.51|53.28|**8.22**|**62.46**|
> |**W4A4KV16 SpinQuant**|5.93|61.98|5.26|65.43|7.38|64.43|13.03|48.98|9.15|57.55|
> |**W4A4KV16 ReSpinQuant**|**5.69**|**62.49**|**5.08**|**65.63**|**7.16**|**65.41**|**12.36**|**49.44**|**8.89**|**59.49**|
>
> These results confirm that ReSpinQuant is effective across a broader range of bit widths. While gains are smaller at higher bit-widths where SpinQuant already nears full precision, ReSpinQuant yields significant improvements under aggressive quantization (e.g., W4A4KV16, W4A4KV4, W3A3KV3) by overcoming the limitations of a single global basis.
>
> Thank you again for the constructive feedback. We hope these additional results address your concerns, and please let us know if you have any remaining concerns.

---

> > ### Author Rebuttal · Reviewer_SqF3 · 2026-04-03
> >
> > Thank you for the rebuttal. The additional quantitative evidence is helpful, and it addresses part of my concerns.
> >
> > However, I still do not find the bit-width justification fully convincing. The rebuttal clarifies the notation and adds two additional settings, but it still does not fully reproduce the broader range of standard `SpinQuant` settings. I also do not find the argument that the method should be most beneficial under aggressive quantization fully sufficient as a substitute for those results. That is a reasonable hypothesis for where gains may be largest, but in my view it is a motivation for prioritizing low-bit settings, not for omitting the more standard ones.
> >
> > Overall, I think the rebuttal makes the paper somewhat stronger, especially on the empirical justification for the low-rank approximation and on bit-width coverage. However, the remaining limitations on evaluation breadth and efficiency generality are still enough that I am keeping my original score.

---

> > > ### Author Response · Authors · 2026-04-07
> > >
> > > Thank you for your response and for continuing to engage with our work. We deeply appreciate your feedback regarding the evaluation breadth.
> > >
> > > We completed the experiments for the remaining W4A8KV16 setting across the LLaMA-2, LLaMA-3, and LLaMA-3.2 families.
> > >
> > > **Additional Results on W4A8KV16:**
> > >
> > > | Method (W4A8KV16) | Llama2-7B (PPL) | Llama2-7B (zeroshot) | Llama2-13B (PPL) | Llama2-13B (zeroshot) | Llama3-8B (PPL) | Llama3-8B (zeroshot) | Llama3.2-1B (PPL) | Llama3.2-1B (zeroshot) | Llama3.2-3B (PPL) | Llama3.2-3B (zeroshot) |
> > > | :--- | :---: | :---: | :---: | :---: | :---: | :---: | :---: | :---: | :---: | :---: |
> > > | **FP** | 5.47 | 65.08 | 4.88 | 67.77 | 6.13 | 68.05 | 9.76 | 55.75 | 7.81 | 63.44 |
> > > | **SpinQuant** | 5.62 | 64.22 | 5.01 | 66.86 | 6.54 | 67.37 | 10.49 | 53.79 | 8.23 | 61.80 |
> > > | **ReSpinQuant** | **5.62** | **64.32** | **4.98** | **67.03** | **6.54** | **67.44** | **10.51** | **53.75** | **8.21** | **62.02** |
> > >
> > > As the table demonstrates, in this less aggressive quantization setting, the performance of all methods is already very close to full-precision. In this regime, ReSpinQuant performs comparably to, or slightly better than, SpinQuant. Overall, we observe notable improvements in the 3-3-3, 4-4-4, and 4-4-16 settings, and comparable performance or slight improvements in the 4-8-8 and 4-8-16 settings.
> > >
> > > Our initial focus on extreme low-bit settings (e.g., W3A3KV3, W4A4KV4) was motivated by the fact that recent methods, such as FlatQuant and OSTQuant, increasingly target these more aggressive 4-bit and 3-bit quantization regimes.
> > >
> > > However, your feedback correctly highlighted that omitting these higher-bit standard settings limits the reader's understanding of our method's overall scope. We completely agree that demonstrating robust performance across both challenging low-bit regimes and standard higher-bit regimes  makes the paper much stronger.
> > >
> > > We will include these comprehensive results covering the full spectrum of bit-widths (including W4A4KV16, W4A8KV16 and W4A8KV8) in the final camera-ready version to ensure a thorough evaluation.
> > >
> > > We hope this new empirical data directly resolves your remaining concerns. Thank you again for helping us improve the completeness of our paper.

---

### Decision · Program_Chairs · 2026-04-30

**Decision:**

Accept (regular)

**Comment:**

This paper proposes ReSpinQuant, a rotation‑based post‑training quantization method for LLMs that learns layer‑wise activation rotations during calibration and fuses most of them into the weights offline. It addresses the gap between global rotation methods (efficient but limited in accuracy because they use one rotation across all layers) and layer‑wise rotation methods (more accurate but costly online because rotations cannot be fused). ReSpinQuant keeps the high accuracy of layer‑wise adaptation while reducing inference overhead by pushing most of the rotation into offline weight fusion and using a lightweight low‑rank subspace correction for the remaining mismatch.

Reviewers appreciated ReSpinQuant’s clear motivation and efficient design, which reconciles the accuracy of layer‑wise rotations with the low overhead of global methods. They highlighted the strong low‑bit results and practical efficiency, including comparisons to recent rotation‑based PTQ methods. The authors added additional results, which mostly addressed reviewers' concerns. The authors are strongly advised to further clarify the presentation in the paper and update the results based on the discussion with the reviewers.